# Offline Goal-Conditioned Reinforcement Learning via $f$-Advantage Regression

**Yecheng Jason Ma, Jason Yan, Dinesh Jayaraman, Osbert Bastani**
University of Pennsylvania
{jasonyma, jasyan, dineshj, obastani}@seas.upenn.edu
https://jasonma2016.github.io/GoFAR/

## Abstract

Offline goal-conditioned reinforcement learning (GCRL) promises general-purpose skill learning in the form of reaching diverse goals from purely offline datasets. We propose **Go**al-conditioned $f$-**A**dvantage **R**egression (GoFAR), a novel regression-based offline GCRL algorithm derived from a state-occupancy matching perspective; the key intuition is that the goal-reaching task can be formulated as a state-occupancy matching problem between a dynamics-abiding imitator agent and an expert agent that directly teleports to the goal. In contrast to prior approaches, GoFAR does not require any hindsight relabeling and enjoys uninterleaved optimization for its value and policy networks. These distinct features confer GoFAR with much better offline performance and stability as well as statistical performance guarantee that is unattainable for prior methods. Furthermore, we demonstrate that GoFAR's training objectives can be re-purposed to learn an agent-independent goal-conditioned planner from purely offline source-domain data, which enables zero-shot transfer to new target domains. Through extensive experiments, we validate GoFAR's effectiveness in various problem settings and tasks, significantly outperforming prior state-of-art. Notably, on a real robotic dexterous manipulation task, while no other method makes meaningful progress, GoFAR acquires complex manipulation behavior that successfully accomplishes diverse goals.

## 1 Introduction

Goal-conditioned reinforcement learning [18, 43, 39] (GCRL) aims to learn a repertoire of skills in the form of reaching distinct goals. *Offline* GCRL [6, 47] is particularly promising because it enables learning general goal-reaching policies from purely offline interaction datasets without any environment interaction [28, 25], which can be expensive in the real-world. As offline datasets contain diverse goals and become increasingly prevalent [9, 19, 6], policies learned this way can acquire a large set of useful primitives for downstream tasks [30]. A central challenge in GCRL is the sparsity of reward signal [3]; without any additional knowledge about the environment, an agent at a state typically only accrues positive binary reward when the state lies within the goal neighborhood. This sparse reward problem is exacerbated in the offline setting, in which the agent cannot explore the environment to discover more informative states about desired goals. Therefore, designing an effective offline GCRL algorithm is a concrete yet challenging path towards general-purpose and scalable policy learning.

In this paper, we present a novel offline GCRL algorithm, **Go**al-conditioned **f-A**dvantage **R**egression (GoFAR), first casting GCRL as a state-occupancy matching [27, 32] problem and then deriving a regression-based policy objective. In particular, GoFAR begins with the following goal-conditioned state-matching objective:

$$\min_\pi \mathrm{D_{KL}}(d^\pi(s;g)\|p(s;g)) \tag{1}$$

36th Conference on Neural Information Processing Systems (NeurIPS 2022).

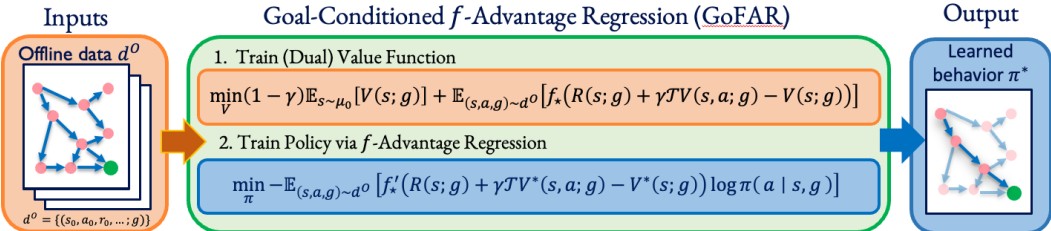

Figure 1: GoFAR schematic illustration.

where $d^\pi(s; g)$ is the goal-conditioned state-occupancy distribution of policy $\pi$ and $p(s; g)$ is the distribution of states that satisfy a particular goal $g$. This objective casts goal-conditioned offline RL as an imitation learning problem: a dynamics-abiding agent imitates as well as possible an expert who can teleport to the goal in one step; see Figure 2. Posing GCRL this way is mathematically principled, as we show that this objective is equivalent to a probabilistic interpretation of GCRL that additionally encourages maximizing state entropy. More importantly, this objectives enables us to extend a state-of-art offline imitation learning algorithm [32] to the goal-conditioned setting and admits elegant optimization using purely offline data by considering an $f$-divergence regularized objective and exploiting ideas from convex duality theory [41, 4]. In particular, we obtain the dual optimal value function $V^*$ from a single unconstrained optimization problem, using which we construct the optimal advantage function (we refer to this as *f-advantage*) that serves as importance weighting for a regression-based policy training objective; see Figure 1 for a schematic illustration.

There are several distinct features to our approach. First and foremost is GoFAR's lack of goal relabeling. Hindsight goal relabeling [3] (known as HER in the literature) relabels trajectory goals to be states that were actually achieved instead of the originally commanded goals. This heuristic is critical for alleviating the sparse-reward problem in prior GCRL methods,

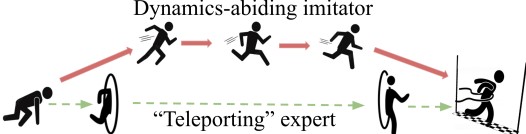

Figure 2: GCRL can be thought of imitating an expert agent that can teleport to goals.

but is unnecessary for GoFAR. GoFAR updates its policy as if all data is already coming from the *optimal* goal-conditioned policy and therefore does not need to perform any hindsight goal relabeling. GoFAR's *relabeling-free* training is of significant practical benefits. First, it enables more stable and simpler training by avoiding sensitive hyperparameter tuning associated with HER that cannot be easily performed offline [48]. Second, hindsight relabeling suffers from hindsight bias in stochastic environments [44], as achieved goals may be accomplished due to noise; this is exacerbated in the offline setting due to inability to collect more data. By bypassing hindsight relabeling, GoFAR promises to be more robust and scalable.

Second, as suggested in Figure 1, GoFAR's value and policy training steps are completely *uninterleaved*: we do not need to update the policy until the value function has converged. This confers greater offline training stability [23, 32] compared to prior works, which mostly involve alternating updates to a critic Q-network and policy network. Furthermore, it enables an algorithmic reduction of GoFAR to weighted regression [7], which allows us to obtain strong finite-sample statistical guarantee on GoFAR's performance; prior regression-based GCRL approaches [14, 47] do not enjoy this reduction and obtain much weaker theoretical guarantees.

Finally, we show that GoFAR can also be used to learn a goal-conditioned *planner* [36, 5]. The key insight lies in observing that, under a mild assumption, GoFAR's dual value function objective does not depend on the action and can learn from state-only offline data. This enables learning a *goal-centric* value function and thereby a near-optimal goal-conditioned planner that is capable of zero-shot transferring to new domains of the same task that shares the goal space. In our experiments, we illustrate that GoFAR planner can indeed plan effective subgoals in a new target domain and enable a low-level controller to reach distant goals that it is not designed for.

We extensively evaluate GoFAR on a variety of offline GCRL environments of varying task complexity and dataset composition, and show that it outperforms all baselines in all settings. Notably, GoFAR is more robust to stochastic environments than methods that depend on hindsight relabeling. We additionally demonstrate that GoFAR learns complex manipulation behavior on a real-world high-dimensional dexterous manipulation task [2], for which most baselines fail to make any progress.

Finally, we showcase GoFAR's planning capability in a cross-robot zero-shot transfer task. To summarize, our contributions are:

1. GoFAR, a novel offline GCRL algorithm derived from goal-conditioned state-matching
2. Detailed technical derivation and analysis of GoFAR's distinct features: (1) relabeling-free, (2) uninterleaved optimization, and (3) planning capability
3. Extensive experimental evaluation of GoFAR, validating its empirical gains and capabilities

## 2 Related Work

**Offline Goal-Conditioned Reinforcement Learning.** One core challenge of (offline) GCRL [18, 43] is the sparse-reward nature of goal-reaching tasks. A popular strategy is hindsight goal relabeling [3] (or HER), which relabels off-policy transitions with future goals they have achieved instead of the original commanded goals. Existing offline GCRL algorithms [6, 47] adapt HER-based online GCRL algorithms to the offline setting by incorporating additional components conducive to offline training. [6] builds on actor-critic style GCRL algorithms [3] by adding conservative Q-learning [24] as well as goal-chaining, which expands the pool of candidate relabeling goals to the entire offline dataset.

Besides actor-critic methods, goal-conditioned behavior cloning, coupled with HER, has been shown to be a simple and effective method [14]. [47] improves upon [14] by incorporating discount-factor and advantage function weighting [37, 38], and shows improved performance in offline GCRL. Diverging from existing literature, GoFAR does not tackle offline GCRL by adapting an online algorithm; instead, it approaches offline GCRL from a novel perspective of state-occupancy matching and derives an elegant algorithm from the dual formulation that is naturally suited for the offline setting and carries many distinct properties that make it more stable, scalable, and versatile.

**State-Occupancy Matching.** State occupancy matching objectives have been shown to be effective for learning from observations [32, 49], exploration [27], as well as matching a hand-designed target state distribution [13]; occupancy matching, in general, has been explored in the imitation learning literature [17, 45, 13, 20]. Our strategy of estimating the state-occupancy ratio is related to DICE techniques for offline policy optimization [33, 35, 26, 21, 32]. The closest work to ours is [32], which proposed $f$-divergence state-occupancy matching for offline imitation learning. Our work shows that such a state-occupancy matching approach can be applied to GCRL. Algorithmically, GoFAR differs from [32] in that it does not require a separate dataset of expert demonstrations; it achieves this by modifying the discriminator, and in some settings can even be used without a discriminator, whereas a discriminator is always required by [32]. Furthermore, we derive several new algorithmic properties (Section 4.2-4.4) that are novel and particularly advantageous for offline GCRL.

## 3 Problem Formulation

**Goal-Conditioned Reinforcement Learning.** We consider an infinite-horizon Markov decision process (MDP) [40] $\mathcal{M} = (S, A, R, T, \mu_0, \gamma)$ with state space $S$, action space $A$, deterministic rewards $r(s, a)$, stochastic transitions $s' \sim T(s, a)$, initial state distribution $\mu_0(s)$, and discount factor $\gamma \in (0, 1]$. A policy $\pi : S \to \Delta(A)$ outputs a distribution over actions to use in a given state. In goal-conditioned RL, the MDP additionally assumes a goal space $G := \{\phi(s) \mid s \in S\}$, where the state-to-goal mapping $\phi : S \to G$ is known. Now, the reward function $r(s; g)$[1] as well as the policy $\pi(a \mid s, g)$ depend on the commanded goal $g \in G$. Given a distribution over desired goals $p(g)$, the objective of goal-conditioned RL is to find a policy $\pi$ that maximizes the discounted return:

$$J(\pi) := \mathbb{E}_{g \sim p(g), s_0 \sim \mu_0, a_t \sim \pi(\cdot|s_t, g), s_{t+1} \sim T(\cdot|s_t, a_t)} \left[ \sum_{t=0}^{\infty} \gamma^t r(s_t; g) \right] \tag{2}$$

The *goal-conditioned* state-action occupancy distribution $d^\pi(s, a; g) : S \times A \times G \to [0, 1]$ of $\pi$ is

$$d^\pi(s, a; g) := (1 - \gamma) \sum_{t=0}^{\infty} \gamma^t \Pr(s_t = s, a_t = a \mid s_0 \sim \mu_0, a_t \sim \pi(s_t; g), s_{t+1} \sim T(s_t, a_t)) \tag{3}$$

which captures the relative frequency of state-action visitations for a policy $\pi$ conditioned on goal $g$. The state-occupancy distribution then marginalizes over actions: $d^\pi(s; g) = \sum_a d^\pi(s, a; g)$. Then, it

---

[1]In GCRL, the reward function customarily does not depend on action.

follows that $\pi(a \mid s, g) = \frac{d^\pi(s,a;g)}{d^\pi(s;g)}$. A state-action occupancy distribution must satisfy the *Bellman flow constraint* in order for it to be an occupancy distribution for some stationary policy $\pi$:

$$\sum_a d(s,a;g) = (1-\gamma)\mu_0(s) + \gamma \sum_{\tilde{s},\tilde{a}} T(s \mid \tilde{s}, \tilde{a})d(\tilde{s}, \tilde{a}; g), \qquad \forall s \in S, g \in G \qquad (4)$$

We write $d^\pi(s,g) = p(g)d^\pi(s;g)$ as the joint goal-state density induced by $p(g)$ and the policy $\pi$. Finally, given $d^\pi$, we can express the objective function (2) as $J(\pi) = \frac{1}{1-\gamma}\mathbb{E}_{(s,g)\sim d^\pi(s,g)}[r(s;g)]$.

**Offline GCRL.** In offline GCRL, the agent cannot interact with the environment $\mathcal{M}$; instead, it is equipped with a static dataset of logged transitions $\mathcal{D} := \{\tau_i\}_{i=1}^N$, where each trajectory $\tau^{(i)} = (s_0^{(i)}, a_0^{(i)}, r_0^{(i)}, s_1^{(i)}, ...; g^{(i)})$ with $s_0^{(i)} \sim \mu_0$ and $g^{(i)}$ is the commanded goal of the trajectory. Note that trajectories need not to be generated from a *goal-directed* agent, in which case $g^{(i)}$ can be randomly drawn from $p(g)$. We denote the empirical goal-conditioned state-action occupancies of $\mathcal{D}^O$ as $d^O(s,a;g)$.

## 4 Goal-Conditioned $f$-Advantage Regression

In this section, we introduce Goal-conditioned $f$-Advantage Regression (GoFAR). We first derive the algorithm in full (Section 4.1), then delve deep into its several appealing properties (Section 4.2-4.4).

### 4.1 Algorithm

We first show that goal-conditioned state-occupancy matching is a mathematically principled approach for solving general GCRL problems, formalizing the teleportation intuition in the introduction.

**Proposition 4.1.** *Given any $r(s;g)$, for each $g$ in the support of $p(g)$, define $p(s;g) = \frac{e^{r(s;g)}}{Z(g)}$, where $Z(g) := \int e^{r(s;g)}ds$ is the normalizing constant. Then, the following equality holds:*

$$-\mathrm{D_{KL}}(d^\pi(s;g)\|p(s;g)) + C = (1-\gamma)J(\pi) + \mathcal{H}(d^\pi(s;g)) \qquad (5)$$

*where $J(\pi)$ is the GCRL objective (Eq. (2)) with reward $r(s;g)$ and $C := \mathbb{E}_{g\sim p(g)}[\log Z(g)]$.*

See Appendix E.1 for proof. This proposition states that, for any choice of reward $r(s;g)$, solving the GCRL problem with a maximum state-entropy regularization is equivalent to optimizing for the goal-conditioned state-occupancy matching objective with target distribution $p(s;g) := \frac{e^{r(s;g)}}{Z(g)}$. Now, the key challenge with optimizing this objective offline is that we cannot sample from $d^\pi(s;g)$. To address this issue, following [32], we first derive an offline lower bound involving an $f$-divergence (see Appendix A for definition) regularization term, which subsequently enables solving this optimization problem via its dual using purely offline data:

**Proposition 4.2.** *Assume[2] for all $g$ in support of $p(g)$, $\forall s, d^O(s;g) > 0$ if $p(s;g) > 0$. Then, for any $f$-divergence that upper bounds the KL-divergence,*

$$-\mathrm{D_{KL}}(d^\pi(s;g)\|p(s;g)) \geq \mathbb{E}_{(s,g)\sim d^\pi(s,g)}\left[\log \frac{p(s;g)}{d^O(s;g)}\right] - \mathrm{D}_f(d^\pi(s,a;g)\|d^O(s,a;g)) \qquad (6)$$

The RHS of (6) can be understood as an $f$-divergence regularized GCRL objective with reward function $R(s;g) = \log \frac{p(s;g)}{d^O(s;g)}$ (we use capital $R$ to differentiate user-chosen reward $R$ from the environment reward $r$). Intuitively, this reward encourages visiting states that occur more often in the "expert" state distribution $p(s;g)$ than in the offline dataset, and the $f$-divergence regularization then ensures that the learned policy is supported by the offline dataset. This choice of reward function can be estimated in practice by training a discriminator [15] $c : S \times G \rightarrow (0,1)$ using the offline data:

$$\min_c \mathbb{E}_{g\sim p(g)}\left[\mathbb{E}_{p(s;g)}[\log c(s,g)] + \mathbb{E}_{d^O(s;g)}[\log 1 - c(s,g)]\right] \qquad (7)$$

We can in fact obtain a looser lower bound that does not require training a discriminator (see B for a derivation):

$$-\mathrm{D_{KL}}(d^\pi(s;g)\|p(s;g)) \geq \mathbb{E}_{(s,g)\sim d^\pi(s,g)}[\log p(s;g)] - \mathrm{D}_f(d^\pi(s,a;g)\|d^O(s,a;g)) \qquad (8)$$

---

[2]This assumption is only needed in our technical derivation to avoid division-by-zero issue.

Because $p(s; g) \propto e^{r(s;g)}$, we may substitute $R(s; g) := r(s; g)$ (when the offline dataset contains reward labels) for $\log p(s; g)$ and bypass having to train a discriminator for reward.

Now, for either choice of lower bound, we may pose the optimization problem with respect to valid choices of state-action occupancies directly, introducing the Bellman flow constraint (4):

$$\max_{d(s,a;g) \geq 0} \quad \mathbb{E}_{(s,g) \sim d(s,g)} [r(s; g)] - \mathrm{D}_f(d(s, a; g) \| d^O(s, a; g))$$

$$\text{(P)} \quad \text{s.t.} \quad \sum_a d(s, a; g) = (1 - \gamma)\mu_0(s) + \gamma \sum_{\tilde{s}, \tilde{a}} T(s \mid \tilde{s}, \tilde{a}) d(\tilde{s}, \tilde{a}; g), \forall s \in S, g \in G \quad (9)$$

This reformulation does not solve the fundamental problem that (9) still requires sampling from $d(s; g)$; however, it has now written the problem in a way amenable to simplification using tools from convex analysis. Now, we show that its dual problem can be reduced to an *unconstrained* minimization problem over the dual variables which serve the role of a value function; importantly, the optimal solution to the dual problem can be used to directly retrieve the optimal primal solution:

**Proposition 4.3.** *The dual problem to* (9) *is*

$$\text{(D)} \quad \min_{V(s,g) \geq 0} (1-\gamma)\mathbb{E}_{(s,g) \sim \mu_0, p(g)}[V(s; g)] + \mathbb{E}_{(s,a,g) \sim d^O} [f_\star (R(s; g) + \gamma\mathcal{T}V(s, a; g) - V(s; g))],$$
$$(10)$$

*where $f_\star$ denotes the convex conjugate function of $f$, $V(s; g)$ is the Lagrangian vector, and $\mathcal{T}V(s, a; g) = \mathbb{E}_{s' \sim T(\cdot|s,a)}[V(s'; g)]$. Given the optimal $V^*$, the primal optimal $d^*$ satisfies:*

$$d^*(s, a; g) = d^O(s, a; g)f'_\star (R(s; g) + \gamma\mathcal{T}V^*(s, a; g) - V^*(s; g)), \forall s \in S, a \in A, g \in G \quad (11)$$

A proof is given in Appendix B. Crucially, as neither expectation in (10) depends on samples from $d$, this objective can be estimated entirely using offline data, making it suitable for offline GCRL. For tabular MDPs, we show that for a suitable choice of $f$-divergence, the optimal $V^*$ in fact admits closed-form solutions; see Appendix D for details. In the continuous control setting, we can optimize (10) using stochastic gradient descent (SGD) by parameterizing $V$ using a deep neural network.

Then, once we have obtained the optimal (resp. converged) $V^*$, we propose learning the policy via the following supervised regression update:

$$\max_\pi \mathbb{E}_{g \sim p(g)} \mathbb{E}_{(s,a) \sim d^O(s,a;g)} [(f'_\star (R(s; g) + \gamma\mathcal{T}V^*(s, a; g) - V^*(s; g)) \log \pi(a \mid s, g)] \quad (12)$$

We see that the regression weights are the first-order derivatives of the convex conjugate of $f$ evaluated at the *dual optimal advantage*, $R(s; g) + \gamma\mathcal{T}V^*(s, a; g) - V^*(s; g)$; we refer to this weighting term as $f$-*advantage*. Hence, we name our overall method *Goal-conditioned $f$-Advantage Regression (GoFAR)*; an abbreviated high-level pseudocode is provided in Algorithm 1 and a detailed version is provided in Appendix C. In practice, we implement GoFAR with $\chi^2$-divergence, a choice of $f$-divergence that is stable for off-policy optimization [49, 32, 26]; see Appendix C for details.

---

**Algorithm 1** Goal-Conditioned $f$-Advantage Regression (Abbreviated); 3-disjoint steps

---

1: (Optionally) Train discriminator-based reward (7)
2: Train (optimal) dual value function $V^*(s; g)$ (10)
3: Train policy $\pi$ via $f$-Advantage Regression (12)

---

Note that (12) forgos directly minimizing (1) offline, which has been found to suffer from training instability [21]. Instead, it naturally incorporates the primal-dual optimal solutions in a regression loss. Now, we will show that this policy objective has several theoretical and practical benefits for offline GCRL that make it particularly appealing.

## 4.2 Optimal Goal-Weighting Property

We show that optimizing (12) automatically obtains the *optimal goal-weighting distribution*. That is, GoFAR trains its policy as if all the data is coming from the optimal goal-conditioned policy for (9). In particular, this property is achieved without any explicit hindsight relabeling (see Appendix A for a technical definition), a mechanism that prior works heavily depend on. To this end, we first define $p(g \mid s, a)$ to be the conditional distribution of goals in the offline dataset conditioned on state-action pair $(s, a)$. Then, according to Bayes rule, we have that

$$p(g \mid s, a) = \frac{d^O(s, a; g)p(g)}{d^O(s, a)} \Rightarrow p(g)d^O(s, a; g) = p(g \mid s, a)d^O(s, a) \quad (13)$$

Using this equality, we can rewrite the policy objective (12) as follows:

$$\min_{\pi} -\mathbb{E}_{(s,a)\sim d^O(s,a)}\mathbb{E}_{g\sim p(g|s,a)}\left[(f'_\star(R(s;g)+\gamma\mathcal{T}V^*(s,a;g)-V^*(s;g))\log\pi(a\mid s,g)\right] \quad (14)$$

$$= \min_{\pi} -\mathbb{E}_{(s,a)\sim d^O(s,a)}\mathbb{E}_{g\sim \tilde{p}(g|s,a)}\left[\log\pi(a\mid s,g)\right] \quad (15)$$

where

$$\tilde{p}(g\mid s,a)\propto p(g\mid s,a)\left(f'_\star(R(s;g)+\gamma\mathcal{T}V^*(s,a;g)-V^*(s;g))\right)=p(g\mid s,a)\frac{d^*(s,a;g)}{d^O(s,a;g)} \quad (16)$$

Thus, we see that GoFAR's $f$-advantage weighting scheme is equivalent to performing supervised policy regression where goals are sampled from $\tilde{p}(g\mid s,a)$. Now, combining (16) and Bayes rule gives

$$d^O(s,a)\tilde{p}(g\mid s,a)\propto d^O(s,a)p(g\mid s,a)\frac{d^*(s,a;g)}{\frac{p(g|s,a)d^O(s,a)}{p(g)}}=p(g)d^*(s,a;g) \quad (17)$$

Thus, we can replace the nested expectations in (15) and obtain that GoFAR policy update amounts to supervised regression of the state-action occupancy distribution of the *optimal* policy to the regularized GCRL problem (9):

$$\pi_{\text{GoFAR}}=\min_{\pi}-\mathbb{E}_{g\sim p(g)}\mathbb{E}_{(s,a)\sim d^*(s,a;g)}\left[\log\pi(a\mid s,g)\right] \quad (18)$$

This derivation makes clear GoFAR's connection with the hindsight goal relabeling mechanism [3] that is ubiquitous in GCRL: GoFAR automatically performs the optimal goal-weighting policy update without any explicit goal relabeling. Furthermore, our derivation also suggests why hindsight relabeling is sub-optimal without further assumption on the reward function [10]: it heuristically chooses $\tilde{p}(g\mid s,a)$ to be the empirical trajectory-wise future achieved goal distribution (i.e., HER), which generally does not coincide with the goals that the optimal policy would reach; see Appendix E.2 for further discussion.

### 4.3 Uninterleaved Optimization and Performance Guarantee

An additional algorithmic advantage of GoFAR is that it *disentangles* the optimization of the value network and the policy network. This can be observed by noting that GoFAR's advantage term (11) is computed using $V^*$, the *optimal* solution of the dual problem. This has the practical significance of disentangling the value-function update (10) from the policy update (12), as we do not need to optimize the latter until the former has converged. This disentanglement is in sharp contrast to prior GCRL works [47, 6, 11, 3], which typically involve alternating updates to the critic Q-network and the policy network, a training procedure that has found to be unstable in the offline setting [23]. This is unavoidable for prior works because their advantage functions are estimated using the Q-estimate of the *current* policy, whereas our advantage term naturally falls out from primal-dual optimality.

The uninterleaved and relabeling-free nature of GoFAR also allows us to derive strong performance guarantees. Because $V^*$ is fixed in (12), this policy objective amounts to a *weighted* supervised learning problem. Therefore, we can extend and adapt mature theoretical results for analyzing Behavior Cloning [1, 42, 46] as well as finite sample error guarantees for weighted regression [7] to obtain statistical guarantees on GoFAR's performance with respect to the optimal $\pi^*$ for (9):

**Theorem 4.1.** *Assume* $\sup_{s,a,g}\frac{d^*(s,a;g)}{d^O(s,a;g)}\leq M$ *and* $\sup|r(s,g)|\leq R_{\max}$. *Consider a policy class* $\Pi:\{S\rightarrow\Delta(A)\}$ *such that* $\pi^*\in\Pi$. *Then, for any* $\delta\in(0,1]$, *with probability at least* $1-\delta$, *GoFAR (18) will return a policy* $\hat{\pi}$ *such that:*

$$V^*-V^{\hat{\pi}}\leq\frac{2R_{\max}M}{(1-\gamma)^2}\sqrt{\frac{\ln(|\Pi|/\delta)}{N}} \quad (19)$$

Notably, the error shrinks as the size of the *offline* data $N$ increases, requiring no dependency on access to data from the "expert" distribution $d^*$. This provides a theoretical basis for GoFAR's empirical scalability as we are guaranteed to obtain good results when the offline data becomes more expansive. In contrast, prior regression-based GCRL methods cannot be easily reduced to a simple weighted regression with respect to the desired goal distribution $p(g)$, so they only obtain weaker results under stronger assumptions on the offline data (e.g, full state-space coverage) as well as the policy; see Appendix E.3 for discussion.

Table 1: Discounted Return, averaged over 10 seeds.

| Task | Supervised Learning | | | Actor-Critic | |
|---|---|---|---|---|---|
| | **GoFAR** (Ours) | **WGCSL** | **GCSL** | **AM** | **DDPG** |
| FetchReach | $28.2 \pm 0.61$ | $21.9 \pm 2.13$ (1.0) | $20.91 \pm 2.78$ (1.0) | $\mathbf{30.1} \pm 0.32$ (0.5) | $29.8 \pm 0.59$ (0.2) |
| FetchPick | $\mathbf{19.7} \pm 2.57$ | $9.84 \pm 2.58$ (1.0) | $8.94 \pm 3.09$ (1.0) | $18.4 \pm 3.51$ (0.5) | $16.8 \pm 3.10$ (0.5) |
| FetchPush $(\star)$ | $\mathbf{18.2} \pm 3.00$ | $14.7 \pm 2.65$ (1.0) | $13.4 \pm 3.02$ (1.0) | $14.0 \pm 2.81$ (0.5) | $12.5 \pm 4.93$ (0.5) |
| FetchSlide | $2.47 \pm 1.44$ | $\mathbf{2.73} \pm 1.64$ (1.0) | $1.75 \pm 1.3$ (1.0) | $1.46 \pm 1.38$ (0.5) | $1.08 \pm 1.35$ (0.5) |
| HandReach $(\star)$ | $\mathbf{11.5} \pm 5.26$ | $5.97 \pm 4.81$ (1.0) | $1.37 \pm 2.21$ (1.0) | $0. \pm 0.0$ (0.5) | $0.81 \pm 1.73$ (0.5) |
| D'ClawTurn $(\star)$ | $\mathbf{9.34} \pm 3.15$ | $0.0 \pm 0.0$ (1.0) | $0.0 \pm 0.0$ (1.0) | $2.82 \pm 1.71$ (1.0) | $0.0 \pm 0.0$ (0.2) |
| Average Rank | **1.5** | 3 | 4.17 | 2.83 | 4 |

### 4.4 Goal-Conditioned Planning

Next, we show that GoFAR can be used to learn a goal-conditioned planner that supports zero-shot transfer to other domains of the same task. The key insight is that GoFAR's value function objective (10) does not depend on actions assuming deterministic transitions. This is because given a transition $(s, a, s', g) \sim d^O$, $\mathcal{T}V(s, a; g) = V(s'; g)$, so we can rewrite the second term in (10) as $\mathbb{E}_{(s,a,s',g)\sim d^O} [f_\star (R(s; g) + \gamma V(s'; g) - V(s; g))]$, which does not depend on the action. This property enables us to learn a *goal-centric* value function that is independent of the agent's state space because the agent's actions are not relevant. Specifically, we propose the following objective:

$$\min_{V(\phi(s),g)\geq 0} (1-\gamma)\mathbb{E}_{(s,g)\sim \mu_0, p(g)}[V(\phi(s); g)]+\mathbb{E}_{(s,s',g)\sim d^O} [f_\star (R(s; g) + \gamma V(\phi(s'), g) - V(\phi(s), g))]$$
(20)

We can think of this objective as learning a value function with the inductive bias that the first operation transforms the state input to the goal space via $\phi$. Since $V$ is now independent of the agent, we can use $f$-advantage regression to instead learn a *goal-conditioned planner*:

$$\max_{\pi} \mathbb{E}_{g\sim p(g)}\mathbb{E}_{(s,s',g)\sim d^O} [(f'_\star(R(s; g) + \gamma V^*(\phi(s'); g) - V^*(\phi(s); g)) \log \pi(\phi(s') \mid \phi(s), g)]$$
(21)

where $\pi$ now outputs the next subgoal $\phi(s')$ conditioned on the current achieved goal $\phi(s)$ and the desired goal $g$. In our experiments, we show how this planner can achieve hierarchical control through zero-shot transfer subgoal plans to a new target domain.

## 5 Experiments

We pose the following questions and provide affirmative answers in our experiments:

1. Is GoFAR effective for offline GCRL? What components are important for performance?
2. Is GoFAR more robust to stochastic environments than hindsight relabeling methods?
3. Can GoFAR be applied to a real-robotics system?
4. Can GoFAR learn a goal-conditioned planner for zero-shot cross-embodiment transfer?

### 5.1 Offline GCRL

**Tasks.** In our simulation experiments, we consider six distinct environments. They include four robot manipulation environments[39]: FetchReach, FetchPickAndPlace, FetchPush, and FetchSlide, and two dexterous manipulation environments: HandReach [39], and D'ClawTurn [2]. All tasks use sparse reward, and their respective goal distributions are de-

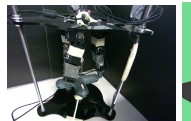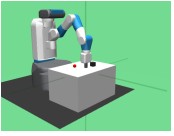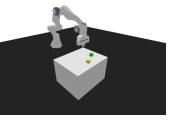

Figure 3: D'Claw (left); Cross-Embodiment transfer source (middle) and target (right) domains.

fined over valid configurations in either the robot space or object space, depending on whether the task involves object manipulation. The offline dataset for each task is collected by either a random policy or a mixture of 90% random policy and 10% expert policy [32, 21], depending on whether random data provides enough coverage of the desired goal distribution. The first five tasks and their datasets are taken from [47]. See Appendix G.3 for dataset details and Appendix F for detailed task descriptions and figures.

**Algorithms.** We compare to state-of-art offline GCRL algorithms, consisting of both regression-based and actor-critic methods. The regression-based methods are: (1) **GCSL** [14], which incorporates hindsight relabeling in conjunction with behavior cloning to clone actions that lead to a specified goal, and (2) **WGCSL** [47], which improves upon GCSL by incorporating discount factor and advantage weighting into the supervised policy learning update. The actor-critic methods are (1) **DDPG** [3], which adapts DDPG [29] to the goal-conditioned setting by incorporating hindsight relabeling, and (2) **ActionableModel (AM)** [6], which incorporates conservative Q-Learning [24] as well as goal-chaining on top of an actor-critic method.

We use tuned hyperparameters for each baseline on all tasks; in particular, we search the best HER ratio from $\{0.2, 0.5, 1.0\}$ for each baseline on each task separately and report the best-performing one. For GoFAR, we use identical hyperparameters as WGCSL for the shared network components and do not tune further. We train each method for 10 seeds, and each training run uses 400k minibatch updates of size 512. Complete architecture and hyperparameter table as well as additional training details are provided in Appendix G.

**Evaluations and Results.** We report the **discounted return** using the sparse binary task reward. This metric rewards algorithms that reach goals as *fast* as possible and stay in the goal region thereafter. In Appendix H, we also report the final **success rate** using the same sparse binary criterion. Because the task reward is binary, these metrics do not take account into how *precisely* a goal is being reached. Therefore, we additionally report the **final distance** to goal in Appendix H.

The full discounted return results are shown in Table 1; ($\star$) indicates statistically significant improvement over the second best method under a two-sample $t$-test. The fraction inside () indicates the best HER rate for each baseline. As shown, GoFAR attains the best overall performance across six tasks, and the results are statistically significant on three tasks, including the two more challenging dexterous manipulation tasks. In Table 6 in Appendix H, we find GoFAR is also superior on the final distance metric. In other words, GoFAR reaches the goals fastest and most precisely.

**Ablations.** Recall that GoFAR achieves optimal goal-weighting, and does not require the HER heuristic that has been key to prior GCRL approaches. We now experimentally confirm that the baselines are not performant without HER. Conversely, we investigate whether HER can help GoFAR. Additionally, we evaluate GoFAR using the sparse binary reward, which implements the looser lower bound in (8).

Finally, we replace $\chi^2$-divergence with KL to understand whether the $f$-divergence lower bound in (6) is necessary. The results are shown in Figure 4 (task-specific breakdown is included in Appendix H). Compared to their original performances (colored dash lines), baselines without HER suffer severely, especially GCSL and WGCSL: both these methods in fact require $100\%$ relabeling and rely solely on HER for positive learning signals. In sharp contrast, GoFAR is also a regression-based method, yet it performs very well without HER and its performance is unaffected by

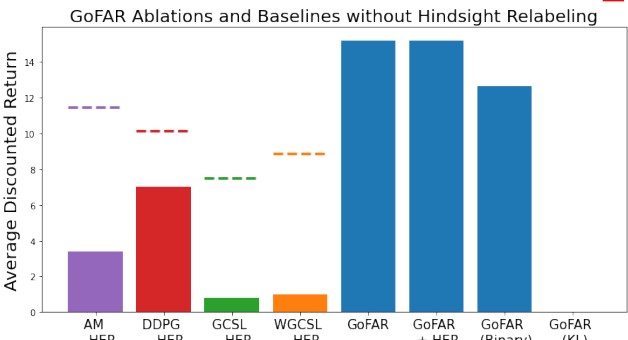

Figure 4: Offline GCRL ablation studies. While GoFAR is robust to hindsight relabeling, removing it is highly detrimental to all baselines.

adding HER. These findings are consistent with the theoretical results of Section 4.2. GoFAR (Binary) performs slightly worse than GoFAR, which is to be expected due to the looser bound; however, it is still better than all baselines, which use the same binary reward and are aided by HER. Again, this ablation highlights the merit of our optimization approach. Finally, GoFAR (KL) suffers from numerical instability and fails to learn, showing that our general $f$-divergence lower bound (6) is not only of theoretical value but also practically significant.

### 5.2 Robustness in Stochastic Offline GCRL Settings

For this experiment, we create noisy variants of the FetchReach environment by adding varying levels of white Gaussian noise to policy action outputs. FetchReach is well-suited for this experiment because all baselines attain satisfactory performance and even outperform GoFAR in Table 1. Further-

more, to isolate the effect of noisy actions, we use GoFAR with binary reward so that all methods now use the same reward input and perform comparatively without added noise. We consider zero-mean Gaussian noise of $\sigma$ value from $0.5, 1$, and $1.5$. For each noise level, we re-collect offline data by executing random actions in the respective noisy environment and train on these noisy offline random data.

The average discounted returns over different noise levels are illustrated in Figure 5. As shown, at noise level $0.5$, while GoFAR's performance is not effected, all baselines already exhibit degraded performance. In particular, DDPG sharply degrades, underperforming GoFAR despite better original performance. At $1.0$, the gap continues to widen, and AM notably collapses, highlighting its sensitivity to noise. This is expected as AM's relabeling mechanism also samples future goals from other trajectories and labels the selected goals with their current Q-value estimates; this procedure becomes highly

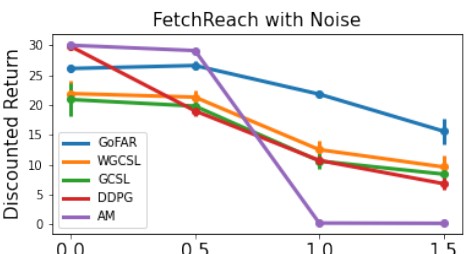

Figure 5: Stochastic environment evaluation. Go-FAR is more robust to stochastic environments due to its lack of hindsight goal relabeling.

noisy when the dataset already exhibits high stochasticity, thus contributing to the instability observed in this experiment. It is only at noise level $1.5$ that GoFAR's performance degradation becomes comparable to those of other methods. Thus, GoFAR's relabeling-free optimization indeed confers greater robustness to environmental stochasticity; in contrast, all baselines suffer from "false positive" relabeled goals due to such disturbances.

## 5.3 Real-World Robotic Dexterous Manipulation

Now, we evaluate on a real D'Claw [2] dexterous manipulation robot (Figure 3 (left)). The task is to have the tri-finger robot rotate the valve to a specified goal angle from its initial angle; see Appendix F for detail. The offline dataset contains 400K transitions, collected by executing purely random actions in the environment, which provides sufficient coverage of the goal space. Compared to the simulated D'Claw, an additional challenge is the inherent stochasticity on a real robot system due to imperfect actuation and hardware wear and tear. Given our conclusion in the stochastic environment experiment above, we should expect GoFAR to perform better.

The training curve of final distances averaged over 3 seeds is shown in Figure 6. We see that GoFAR exhibits smooth progress over training and significantly outperforms all baselines. At the end of training, GoFAR is able to rotate the valve on average within **10** degrees of the specified goal angle. In contrast, most baselines do not make any progress on this difficult task and learns policies that do not manage to turn the valve at all. AM is the only baseline that learns a non-trivial policy; however, its training fluctuates significantly, and the learned policy often executes extreme actions during evaluation

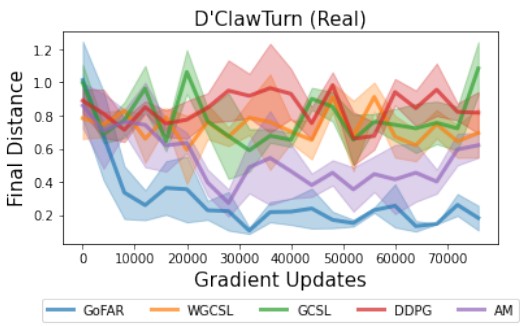

Figure 6: Real-robot results.

that fail to reach the desired goal. We provide qualitative analysis in Appendix H.3, and videos are included in the supplement.

## 5.4 Zero-Shot Transfer Across Robots

For this experiment, we consider zero-shot transfer from the FetchPush task (source domain) to an equivalent planar object pushing task using the 7-DOF Franka Panda robot [12]; the environments are illustrated in Figure 3. In the target domain, we assume a pre-programmed or learned low-level goal-reaching controller $\pi_{\text{low}}(a \mid s, g)$

Table 2: Zero-shot transfer results.

| Method | Success Rate |
|---|---|
| GoFAR Planner (Oracle) | 84% |
| GoFAR Hierarchical Controller | 37% |
| Low-Level Controller | 19% |

that can push objects over short distances but fails for longer tasks. For these distant goals, we train the goal-conditioned GoFAR value function $V^*(\phi(s); g)$ and planner $\pi(\phi(s') \mid \phi(s), g)$ according to (20) and (21), to set goals for $\pi_{\text{low}}$. We use the same offline FetchPush data as in Section 5.1. We compare two approaches: (1) naively using the **Low-Level Controller**, and (2) using the GoFAR planner to guide the low-level controller (**GoFAR Hierarchical Controller**). See Appendix G.4 for additional details. We report the success rate over 100 random distant test goals in Table 2. We see that naively commanding the low-level controller with these distant goals indeed results in very low success rate. When the controller is augmented with GoFAR planner, the success rate nearly doubles. Note that our planner itself is much superior: if all subgoals are perfectly reached, **GoFAR Planner (Oracle)** achieves 84% success. See Appendix G.4 for additional experimental details. These experiments validate GoFAR's ability to zero-shot transferring subgoal plans to enhance the capabability of low-level controllers. Note that we have no access to any data from the target domain, and this planning capability naturally emerges from our training objectives and does not require any change in the algorithm. We show qualitative results in Appendix H.4 and videos in the supplement.

## 6 Conclusion

We have presented GoFAR, a novel regression-based offline GCRL algorithm derived from a state-occupancy matching perspective. GoFAR is relabeling-free and enjoys uninterleaved training, making it both theoretically and practically advantageous compared to prior state-of-art. Furthermore, GoFAR supports training a goal-conditioned planner with promising zero-shot transfer capability. Through extensive experiments, we have validated the practical utility of GoFAR in various challenging problem settings, showing significant improvement over prior state-of-art. We believe that GoFAR's strong theoretical foundation and empirical performance make it an ideal candidate for scalable skill learning in the real world.

## Acknowlegement

We thank Edward Hu and Oleh Rybkin for their feedback on an earlier draft of the paper, Kun Huang for assistance on the real-robot experiment setup, and Selina Li for assistance on graphic design. This work is funded in part by an Amazon Research Award, gift funding from NEC Laboratories America, NSF Award CCF-1910769, NSF Award CCF-1917852 and ARO Award W911NF-20-1-0080. The U.S. Government is authorized to reproduce and distribute reprints for Government purposes notwithstanding any copyright notation herein.

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
