# Part I

# Appendix

These appendices contain material supplementary to submission 7863: "Offline Goal-Conditioned Reinforcement Learning via $f$-Advantage Regression"

## Table of Contents

## A   Additional Technical Background

We provide some technical definitions that are needed in our proofs in Appendix B and discussions of hindsight relabeling in Section 4.1 of the main text as well as Appendix E.2.

## A.1 $f$-Divergence and Fenchel Duality

These definitions are adapted from [32, 34].

**Definition A.1** ($f$-divergence). For any continuous, convex function $f$ and two probability distributions $p, q \in \Delta(\mathcal{X})$ over a domain $\mathcal{X}$, the $f$-divergence of $p$ computed at $q$ is defined as

$$D_f(p\|q) = \mathbb{E}_{x \sim q}\left[f\left(\frac{p(x)}{q(x)}\right)\right] \tag{22}$$

Some common choices of $f$-divergence includes the KL-divergence and the $\chi^2$-divergence, which corresponds to choosing $f(x) = x \log x$ and $f(x) = \frac{1}{2}(x-1)^2$, respectively.

**Definition A.2** (Fenchel conjugate). Given a vector space $\mathcal{X}$ with inner-product $\langle \cdot, \cdot \rangle$, the *Fenchel conjugate* $f_\star : \mathcal{X}_\star \to \mathbb{R}$ of a convex and differentiable function $f : \Omega \to \mathbb{R}$ is

$$f_\star(y) := \max_{x \in \mathcal{X}}\langle x, y \rangle - f(x) \tag{23}$$

and any maximizer $x^*$ of $f_\star(y)$ satisfies $x^* = f'_\star(y)$.

For an $f$-divergence, under mild realizability assumptions [8] on $f$, the Fenchel conjugate of $D_f(p\|q)$ at $y : \mathcal{X} \to \mathbb{R}$ is

$$D_{\star,f}(y) = \max_{p \in \Delta(\mathcal{X})} \mathbb{E}_{x \sim p}[y(x)] - D_f(p\|q) \tag{24}$$

$$= \mathbb{E}_{x \sim q}[f_\star(y(x))] \tag{25}$$

and any maximizer $p^*$ of $D_{\star,f}(y)$ satisfies $p^*(x) = q(x)f'_\star(y(x))$. These optimality conditions can be seen as extensions of the KKT-condition.

## A.2 Hindsight Goal-Relabeling

We provide a mathematical formalism of hindsight goal relabeling [3].

**Definition A.3.** Given a state $s_t$ from a trajectory $\tau = \{s_0, a_0, r_0, ..., s_T; g\}$, hindsight goal-relabeling is the goal-relabeling distribution

$$p_{\text{HER}}(g \mid s_t, a_t, \tau) = q[\phi(s_t), ..., \phi(s_T)] \tag{26}$$

where $q$ is some categorical distribution taking values in $\{\phi(s_t), ..., \phi(s_T)\}$.

That is, the relabeled goal is selected from some distribution goals that are reached in the future in the same trajectory. The most canonical choice of $q$, known as *hindsight experience replay* (HER), selects $q$ to be the uniform distribution. Once a goal $\tilde{g}$ is chosen, the reward label is also re-computed using the reward function assumed by the algorithm: $r_t := r(s_t, \tilde{g})$.

# B Proofs

In this section, we restate propositions and theorems in the paper and present their proofs.

## B.1 Proof of Proposition 4.2

**Proposition B.1.** *Assume for all $g$ in support of $p(g)$, $\forall s, d^O(s; g) > 0$ if $p(s; g) > 0$. Then, for any $f$-divergence that upper bounds the KL-divergence,*

$$-D_{\text{KL}}(d^\pi(s; g)\|p(s; g)) \geq \mathbb{E}_{(s,g) \sim d^\pi(s,g)}\left[\log \frac{p(s; g)}{d^O(s; g)}\right] - D_f(d^\pi(s, a; g)\|d^O(s, a; g)) \tag{27}$$

$$\geq \mathbb{E}_{(s,g) \sim d^\pi(s,g)}[\log p(s; g)] - D_f(d^\pi(s, a; g)\|d^O(s, a; g)) \tag{28}$$

*Proof.* We first present and prove some technical lemma needed to prove this result. The following lemmas and proofs are adapted from [32]; in particular, we extend these known results to the goal-conditioned setting.

**Lemma B.1.** *For any pair of valid occupancy distributions $d_1$ and $d_2$, we have*

$$\mathrm{D_{KL}}(d_1(s;g)\|d_2(s;g) \leq \mathrm{D_{KL}}(d_1(s,a;g)\|d_2(s,a;g)) \tag{29}$$

*Proof.* This lemma hinges on proving the following lemma first.

**Lemma B.2.**

$$\mathrm{D_{KL}}(d_1(s,a,s';g)\|d_2(s,a,s';g)) = \mathrm{D_{KL}}(d_1(s,a;g)\|d_2(s,a;g)) \tag{30}$$

*Proof.*

$$\mathrm{D_{KL}}(d_1(s,a,s';g)\|d_2(s,a,s';g))$$
$$= \int_{S \times A \times S \times G} p(g) d_1(s,a,s';g) \log \frac{d_1(s,a;g) \cdot T(s' \mid s,a)}{d_2(s,a;g) \cdot T(s' \mid s,a)} ds' dadsdg$$
$$= \int_{S \times A \times S \times G} p(g) d_1(s,a,s';g) \log \frac{d_1(s,a;g)}{d_2(s,a;g)} ds' dadsdg$$
$$= \int_{S \times A \times G} p(g) d_1(s,a;g) \log \frac{d_1(s,a;g)}{d_2(s,a;g)} dadsdg$$
$$= \mathrm{D_{KL}}(d_1(s,a;g)\|d_2(s,a;g))$$

$\square$

Using this result, we can prove Lemma B.1:

$$\mathrm{D_{KL}}(d_1(s,a;g)\|d_2(s,a;g))$$
$$= \mathrm{D_{KL}}(d_1(s,a,s';g)\|d_2(s,a,s';g))$$
$$= \int_{S \times A \times S \times G} p(g) d_1(s,a,s';g) \log \frac{d_1(s,a;g) \cdot T(s' \mid s,a)}{d_2(s,a;g) \cdot T(s' \mid s,a)} ds' dadsdg$$
$$= \int_{S \times A \times S \times G} p(g) d_1(s;g) \pi_1(a \mid s,g) T(s' \mid s,a) \log \frac{d_1(s,a;g) \cdot T(s' \mid s,a)}{d_2(s,a;g) \cdot T(s' \mid s,a)} ds' dadsdg$$
$$= \int p(g) d_1(s;g) \pi_1(a \mid s,g) T(s' \mid s,a) \log \frac{d_1(s;g)}{d_2(s;g)} ds' dadsdg$$
$$+ \int p(g) d_1(s;g) \pi_1(a \mid s,g) T(s' \mid s,a) \log \frac{\pi_1(a \mid s,g) T(s' \mid s,a)}{\pi_2(a \mid s,g) T(s' \mid s,a)} ds' dadsdg$$
$$= \int p(g) d_1(s;g) \log \frac{d_1(s;g)}{d_2(s;g)} dsdg + \int p(g) d_1(s;g) \pi_1(a \mid s,g) \log \frac{\pi_1(a \mid s,g)}{\pi_2(a \mid s,g)} dadsdg$$
$$= \mathrm{D_{KL}}(d_1(s;g)\|d_2(s;g)) + \mathrm{D_{KL}}(\pi_1(a \mid s,g)\|\pi_2(a \mid s,g))$$
$$\geq \mathrm{D_{KL}}(d_1(s;g)\|d_2(s;g))$$

$\square$

Now given these technical lemmas, we have

$$\mathrm{D_{KL}}(d^\pi(s;g)\|p(s;g))$$
$$= \int p(g) d^\pi(s;g) \log \frac{d^\pi(s;g)}{p(s;g)} \cdot \frac{d^O(s;g)}{d^O(s;g)} dsdg, \quad \text{we assume that } d^O(s;g) > 0 \text{ whenever } p(s;g) > 0.$$
$$= \int p(g) d^\pi(s;g) \log \frac{d^O(s;g)}{p(s;g)} dsdg + \int p(g) d^\pi(s;g) \log \frac{d^\pi(s;g)}{d^O(s;g)} dsdg$$
$$\leq \mathbb{E}_{(s,g) \sim d^\pi(s,g)} \left[ \log \frac{d^O(s;g)}{p(s;g)} \right] + \mathrm{D_{KL}}(d^\pi(s,a;g)\|d^O(s,a;g))$$

where the last step follows from Lemma B.1. Then, for any $\mathrm{D}_f \geq \mathrm{D_{KL}}$, we have that

$$-\mathrm{D_{KL}}(d^\pi(s;g)\|p(s;g)) \geq \mathbb{E}_{(s,g) \sim d^\pi(s,g)} \left[ \log \frac{p(s;g)}{d^O(s;g)} \right] - \mathrm{D}_f(d^\pi(s,a;g)\|d^O(s,a;g)) \tag{31}$$

Then, since $\mathbb{E}_{(s,g)\sim d^\pi(s,g)}\left[\log\frac{1}{d^O(s;g)}\right]\geq 0$, we also obtain the following looser bound:

$$-D_{\mathrm{KL}}\left(d^\pi(s;g)\|p(s;g)\right)\geq \mathbb{E}_{(s,g)\sim d^\pi(s,g)}\left[\log p(s;g)\right]-\mathrm{D}_f\left(d^\pi(s,a;g)\|d^O(s,a;g)\right)\qquad(32)$$

$\square$

## B.2 Proof of Proposition 4.3

**Proposition B.2.** *The dual problem to* (9) *is*

(D) $\quad\displaystyle\min_{V(s,g)\geq 0}(1-\gamma)\mathbb{E}_{(s,g)\sim\mu_0,p(g)}[V(s;g)]+\mathbb{E}_{(s,a,g)\sim d^O}\left[f_\star\left(r(s,g)+\gamma\mathcal{T}V(s,a;g)-V(s;g)\right)\right],$
(33)

*where $f_\star$ denotes the convex conjugate function of $f$, $V(s;g)$ is the Lagrangian vector, and $\mathcal{T}V(s,a;g)=\mathbb{E}_{s'\sim T(\cdot|s,a)}[V(s';g)]$. Given the optimal $V^*$, the primal optimal $d^*$ satisfies:*

$$d^*(s,a;g)=d^O(s,a;g)f'_\star\left(r(s,g)+\gamma\mathcal{T}V^*(s,a,g)-V^*(s,g)\right),\forall s\in S, a\in A, g\in G\qquad(34)$$

*Proof.* We begin by writing the Lagrangian dual of the primal problem:

$$\min_{V(s;g)\geq 0}\max_{d(s,a;g)\geq 0}\mathbb{E}_{(s,g)\sim d(s,g)}\left[\log\left(r(s;g)\right)\right]-\mathrm{D}_f(d(s,a;g)\|d^O(s,a;g))$$

$$+\sum_{s,g}p(g)V(s;g)\left((1-\gamma)\mu_0(s)+\gamma\sum_{\tilde{s},\tilde{a}}T(s\mid\tilde{s},\tilde{a})d(\tilde{s},\tilde{a};g)-\sum_a d(s,a;g)\right)\qquad(35)$$

where $p(g)V(s;g)$ is the Lagrangian vector. Then, we note that

$$\sum_{s,g}V(s;g)\sum_{\tilde{s},\tilde{a}}T(s\mid\tilde{s},\tilde{a})d(\tilde{s},\tilde{a};g)=\sum_{\tilde{s},\tilde{a},g}d(\tilde{s},\tilde{a};g)\sum_s T(s\mid\tilde{s},\tilde{a})V(s;g)=\sum_{s,a,g}d(s,a;g)\mathcal{T}V(s,a;g)$$
(36)

Using this, we can rewrite (35) as

$$\min_{V(s;g)\geq 0}\max_{d(s,a;g)\geq 0}(1-\gamma)\mathbb{E}_{(s,g)\sim(\mu_0,p(g))}[V(s;g)]+\mathbb{E}_{(s,a,g)\sim d}\left[(r(s;g)+\gamma\mathcal{T}V(s,a;g)-V(s;g))\right]$$

$$-\mathrm{D}_f(d(s,a;g)\|d^O(s,a;g))$$
(37)

And finally,

$$\min_{V(s,g)\geq 0}(1-\gamma)\mathbb{E}_{(s,g)\sim(\mu_0,p(g))}[V(s;g)]+\max_{d(s,a;g)\geq 0}\mathbb{E}_{(s,a,g)\sim d}\left[(r(s,g)+\gamma\mathcal{T}V(s,a;g)-V(s;g))\right]$$

$$-\mathrm{D}_f(d(s,a;g)\|d^O(s,a;g))$$
(38)

Now, we make the key observation that the inner maximization problem in (38) is in fact the Fenchel conjugate of $\mathrm{D}_f(d(s,a,g)\|d^O(s,a,g))$ at $r(s,g)+\gamma\mathcal{T}V(s,a,g)-V(s,g)$. Therefore, we can reduce (38) to an unconstrained minimization problem over the dual variables

$$\min_{V(s,g)\geq 0}(1-\gamma)\mathbb{E}_{(s,g)\sim\mu_0,p(g)}[V(s;g)]+\mathbb{E}_{(s,a,g)\sim d^O}\left[f_\star\left(r(s,g)+\gamma\mathcal{T}V(s,a;g)-V(s;g)\right)\right],$$
(39)

and consequently, we can relate the dual-optimal $V^*$ to the primal-optimal $d^*$ using Fenchel duality (see Appendix A)

$$d^*(s,a;g)=d^O(s,a;g)f'_\star\left(r(s,g)+\gamma\mathcal{T}V^*(s,a,g)-V^*(s,g)\right),\forall s\in S, a\in A, g\in G,\qquad(40)$$

as desired. $\square$

## B.3 Proof of Theorem 4.1

**Theorem B.3.** *Assume* $\sup_{s,a,g}\frac{d^*(s,a;g)}{d^O(s,a;g)}\leq M$ *and* $\sup|r(s,g)|\leq R_{\max}$. *Consider a policy class* $\Pi:\{S\to\Delta(A)\}$ *such that* $\pi^*\in\Pi$. *Then, for any* $\delta\in(0,1]$, *with probability at least* $1-\delta$, *GoFAR* (18) *will return a policy* $\hat{\pi}$ *such that:*

$$V^*-V^{\hat{\pi}}\leq\frac{2R_{\max}M}{(1-\gamma)^2}\sqrt{\frac{\ln(|\Pi|/\delta)}{N}}\qquad(41)$$

*where* $V^\pi:=\mathbb{E}_{(s,g)\sim(\mu_0,g)}[V(s;g)]$.

*Proof.* We begin by deriving an upper bound using the performance difference lemma [1]:

$$V^* - V^{\hat{\pi}} \leq \frac{1}{1-\gamma}\mathbb{E}_{s \sim d^*, g \sim p(g)}\mathbb{E}_{a \sim \pi^*(\cdot|s,g)}A^{\hat{\pi}}(s,a;g) \tag{42}$$

Then, using standard algebraic manipulations, we have:

$$\begin{aligned}
&\frac{1}{1-\gamma}\mathbb{E}_{s \sim d^*, g \sim p(g)}\mathbb{E}_{a \sim \pi^*(\cdot|s,g)}A^{\hat{\pi}}(s,a;g) \\
=&\frac{1}{1-\gamma}\mathbb{E}_{s \sim d^*, g \sim p(g)}\left[\mathbb{E}_{a \sim \pi^*(\cdot|s,g)}A^{\hat{\pi}}(s,a;g) - \mathbb{E}_{a \sim \hat{\pi}(\cdot|s,g)}A^{\hat{\pi}}(s,a;g)\right] \\
\leq&\frac{R_{\max}}{(1-\gamma)^2}\mathbb{E}_{s \sim d^*, g \sim p(g)}\left[\|\pi^*(\cdot \mid s,g) - \hat{\pi}(\cdot \mid s,g)\|_1\right] \\
\leq&\frac{2R_{\max}}{(1-\gamma)^2}\mathbb{E}_{s \sim d^*, g \sim p(g)}\left[\|\pi^*(\cdot \mid s,g) - \hat{\pi}(\cdot \mid s,g)\|_{\text{TV}}\right]
\end{aligned} \tag{43}$$

Then, since $\sup_{s,a,g}\frac{d^*(s,a;g)}{d^O(s,a;g)} \leq M$, we can use Hoeffding's inequality with weighted empirical loss [7] to obtain that:

$$V^* - V^{\hat{\pi}} \leq \frac{2R_{\max}M}{(1-\gamma)^2}\sqrt{\frac{\ln(|\Pi|/\delta)}{N}} \tag{44}$$

$\square$

## C  GoFAR Technical Details

In this section, we provide additional technical details of GoFAR that are omitted in the main text. These include (1) detail of the GoFAR discriminator training, (2) mathematical expressions of GoFAR specialized to common $f$-Divergences, and (3) a full pseudocode.

### C.1  Discriminator Training

Training the discriminator 7 in practice requires choosing $p(s;g)$. For simplicity, we set $p(s;g)$ to be the Dirac distribution centered at $g$: $\mathbb{I}(\phi(s) = g)$; this precludes having to choose hyperparameters for $p(s;g)$.

Once the discriminator has converged, we can retrieve the reward function $R(s;g) = \log\frac{p(s;g)}{d^O(s;g)} = -\log\left(\frac{1}{c^*(s;g)} - 1\right)$, since $c^*(s;g) = \frac{d^O(s;g)}{p(s;g)+d^O(s;g)}$.

### C.2  GoFAR with common $f$-Divergences

GoFAR requires choosing a $f$-divergence. Here, we specialize GoFAR to $\chi^2$-divergence as well as KL-divergence as examples. Our practical implementation uses $\chi^2$-divergence, which we found to be significantly more stable than KL-divergence (see Section 5.1).

**Example 1** (GoFAR with $\chi^2$-divergence). $f(x) = \frac{1}{2}(x-1)^2$, and we can show that $f_\star(x) = \frac{1}{2}(x+1)^2$ and $f'_\star(x) = x+1$. Hence, the GoFAR objective amounts to

$$\min_{V(s;g) \geq 0}(1-\gamma)\mathbb{E}_{(s,g) \sim (\mu_0, p(g))}[V(s;g)] + \frac{1}{2}\mathbb{E}_{(s,a,g) \sim d^O}\left[(R(s;g) + \gamma\mathcal{T}V(s,a;g) - V(s;g) + 1)^2\right] \tag{45}$$

and

$$d^*(s,a;g) = d^O(s,a;g)\max\left(0, R(s,a;g) + \gamma\mathcal{T}V^*(s,a;g) - V^*(s;g) + 1\right) \tag{46}$$

**Example 2** (GoFAR with KL-divergence). We have $f(x) = x\log x$ and that $D_{\star,f}(y) = \log\mathbb{E}_{x \sim q}[\exp y(x)]$ [4]. Hence, the KL-divergence GoFAR objective is

$$\min_{V(s;g) \geq 0}(1-\gamma)\mathbb{E}_{(s,g) \sim (\mu_0, p(g))}[V(s;g)] + \log\mathbb{E}_{(s,a,g) \sim d^O}\left[\exp\left(R(s;g) + \gamma\mathcal{T}V(s,a;g) - V(s;g)\right)\right] \tag{47}$$

and

$$d^*(s,a;g) = d^O(s,a;g)\text{softmax}\left(R(s;g) + \gamma\mathcal{T}V^*(s,a;g) - V^*(s;g)\right) \tag{48}$$

Now, we provide the full pseudocode for GoFAR implemented using $\chi^2$-divergence in Algorithm 2.

## C.3 Full Pseudocode

---

**Algorithm 2** GoFAR for Continuous MDPs

---

1: **Require**: Offline dataset $d^O$, choice of $f$-divergence $f$, choice of $p(s;g)$
2: Randomly initialize discriminator $c_\psi$, value function $V_\theta$, and policy $\pi_\phi$.
3: // Train Discriminator (Optional)
4: **for** number of discriminator iterations **do**
5:     Sample minibatch $\{s_d^i, g^i\}_{i=1}^N \sim d^O$
6:     Sample $\{s_g^j\}_{j=1}^M \sim p(s;g^i) \quad \forall i \in 1 \ldots N$
7:     Discriminator objective: $\mathcal{L}_c(\psi) = \frac{1}{N} \sum_{i=1}^N [\log(1 - c_\psi(s_d^j, g^i)) + \frac{1}{M} \sum_{j=1}^M [\log c_\psi(s_g^j, g^i)]]$
8:     Update $c_\psi$ using SGD: $c_\psi \leftarrow c_\psi - \alpha_c \nabla \mathcal{L}_c(\psi)$
9: **end for**
10: // Train Dual Value Function
11: **for** number of value iterations **do**
12:     Sample minibatch of offline data $\{s_t^i, a_t^i, s_{t+1}^i, g_t^i\}_{i=1}^N \sim d^O, \{s_0^i\}_{i=1}^M \sim \mu_0, \{g_0^i\}_{i=1}^M \sim d^O$
13:     If discriminator, obtain reward: $R(s_t^i; g_t^i) = -\log\left(\frac{1}{c_\psi(s_t^i; g_t^i)} - 1\right) \quad \forall i = 1 \ldots N$
14:     If no discriminator, obtain reward: $\{R(s_t^i; g_t^i)\}_{i=1}^N \sim d^O$
15:     Value objective: $\mathcal{L}_V(\theta) = \frac{1-\gamma}{M} \sum_{i=1}^M [V_\theta(s_0^i; g_0^i)] + \frac{1}{N} \sum_{i=1}^N \left[ f_\star(R_t^i + \gamma V(s_{t+1}^i; g_t^i) - V(s_t^i; g_t^i)) \right]$
16:     Update $V_\theta$ using SGD: $V_\theta \leftarrow V_\theta - \alpha_V \nabla \mathcal{L}_V(\theta)$
17: **end for**
18: // Train Policy With $f$-Advantage Regression
19: **for** number of policy iterations **do**
20:     Sample minibatch of offline data $\{s_t^i, a_t^i, s_{t+1}^i, g_t^i\}_{i=1}^N \sim d^O$
21:     If discriminator, obtain reward: $R(s_t^i; g_t^i) = -\log\left(\frac{1}{c_\psi(s_t^i; g_t^i)} - 1\right) \quad \forall i = 1 \ldots N$
22:     If no discriminator, obtain reward: $\{R(s_t^i; g_t^i)\}_{i=1}^N \sim d^O$
23:     Policy objective: $\mathcal{L}_\pi(\phi) = \sum_{i=1}^N \left[ \left( f'_\star \left( R_t^i + \gamma V_\theta(s_{t+1}^i; g_t^i) - V_\theta(s_t^i; g_t^i) \right) \log \pi(a \mid s, g) \right) \right]$
24:     Update $\pi_\phi$ using SGD: $\pi_\phi \leftarrow \pi_\phi - \alpha_\pi \nabla \mathcal{L}(\phi)$
25: **end for**

---

# D GoFAR for Tabular MDPs

In Section 4.1 of the main text, we have stated that in tabular MDPs, GoFAR's optimal dual value function (10) admits closed-form solution when we choose $\chi^2$-divergence. Here, we provide a derivation of this result.

Recall the dual problem (1)

$$\min_{V(s;g) \geq 0} (1-\gamma) \mathbb{E}_{(s,g) \sim (\mu_0, p(g))}[V(s;g)] + \frac{1}{2} \mathbb{E}_{(s,a,g) \sim d^O} \left[ (R(s;g) + \gamma \mathcal{T} V(s,a;g) - V(s;g) + 1)^2 \right] \tag{49}$$

To derive a closed-form solution, we rewrite the problem in vectorized notation; we borrow our notations from [32]. We first augment the state-space by concatenating the state dimensions and the goal dimensions so that the new state space $\tilde{S}$ has dimension $S + G$. Then, the new transition function, with slight abuse of notation, $T((s', g') \mid (s, g), a) = T(s \mid s, a) \mathbb{I}(g' = g)$; the new initial state distribution is thus $\mu_0(s, g) = \mu_0(s) p(g)$. Therefore, $\tilde{T} \in \mathbb{R}_+^{|S||G||A| \times |S||G|}$ and $\mu_0 \in \mathbb{R}_+^{|S||G|}$.

We assume that the offline dataset $\mathcal{D}^O$ is collected by a behavior policy $\pi_b$. We construct a surrogate MDP $\hat{\mathcal{M}}$ using maximum likelihood estimation; that is, $\hat{T}((s', g') \mid (s, g), a) = \frac{n(s,a,s')}{n(s,a)} \mathbb{I}(g' = g)$, and we impute $\hat{T}((s', g) \mid (s, g), a) = \frac{1}{S}$ when $n(s, a) = 0$. Then, using $\hat{\mathcal{M}}$, we can compute $d^O \in \mathbb{R}_+^{|S||G||A|}$ using linear programming and define reward $r \in \mathbb{R}_+^{|S||G|}$ as $r(s; g) = \log \frac{p(s;g)}{d^O(s;g)}$, where $p(s;g) \in \mathbb{R}_+^{|S||G|}$. Now, define $\mathcal{T} \in \mathbb{R}^{|S||G||A| \times |S||G|}$ such that $(\mathcal{T} V)(s, a; g) = \sum s' T((s', g) \mid (s, g), a) V(s'; g)$, where $V \in \mathbb{R}_+^{|S||G|}$ is the dual optimization variables. We also define $\mathcal{B} \in \mathbb{R}^{|S||G||A| \times |S||G|}$ such that $(\mathcal{B} V)(s, a; g) = V(s; g)$. Finally, we define

$D = \text{diag}(d^O) \in \mathbb{R}^{|S||G||A| \times |S||G||A|}$. Now, we can rewrite the dual problem as follow:

$$\min_{V(s;g) \geq 0} (1-\gamma)\mathbb{E}_{(s,g)\sim(\mu_0,p(g))}[V(s;g)] + \frac{1}{2}\mathbb{E}_{(s,a,g)\sim d^O}\left[(R(s;g) + \gamma\mathcal{T}V(s,a;g) - V(s;g) + 1)^2\right]$$

$$\Rightarrow \min_{V(s;g)} (1-\gamma)\mu_0^\top V + \frac{1}{2}\mathbb{E}_{(s,a,G)\sim d^O}\left[\left(\underbrace{\mathcal{B}R(s,a;g) + \gamma\mathcal{T}V(s,a;g) - \mathcal{B}V(s,a;g)}_{R_V(s,a;g)} + 1\right)^2\right]$$

$$\Rightarrow \min_{V(s;g)} (1-\gamma)\mu_0^\top V + \frac{1}{2}(R_V + I)^\top D(R_V + I)$$

$$(50)$$

Now, we recognize that (50) is equivalent to Equation 49 in [32], as we have reduced goal-conditioned RL to regular RL with an augmented state-space. Now, using the same derivation as in [32], we have that

$$V^* = \left((\gamma\mathcal{T} - \mathcal{B})^\top D(\gamma\mathcal{T} - \mathcal{B})\right)^{-1}\left((\gamma - 1)\mu_0 + (\mathcal{B} - \gamma\mathcal{T})^\top D(I + BR)\right) \quad (51)$$

and we can recover $d^*(s, a; g)$:

$$d^*(s, a; g) = d^O(s, a; g)\left(\mathcal{B}R(s,a;g) + \gamma\mathcal{T}V^*(s,a;g) - \mathcal{B}V^*(s,a;g) + 1\right) \quad (52)$$

Given $d^*$, we may extract the optimal policy $\pi^*$ by marginalizing over actions:

$$\pi^*(a \mid s, g) = \frac{d^*(s, a; g)}{\sum_a d^*(s, a'; g)} = \frac{d^O(s, a; g)(R(s;g) + \gamma\mathcal{T}V(s,a;g) - V(s;g))}{\sum_a d^O(s, a; g)(R(s;g) + \gamma\mathcal{T}V(s,a;g) - V(s;g))} \quad (53)$$

# E    Additional Technical Discussion

## E.1    Connecting Goal-Conditioned State-Matching and Probabilistic GCRL

Suppose the GCRL problem comes with a reward function $r(s; g)$. We also show that there is an equivalent goal-conditioned state-occupancy matching problem with a target distribution $p(s; g)$ defined based on $r(s; g)$.

**Proposition E.1.** *(Proposition 4.1 in the paper) Given any $r(s; g)$, for each $g$ in the support of $p(g)$, define $p(s; g) = \frac{e^{r(s;g)}}{Z(g)}$, where $Z(g) := \int e^{r(s;g)}ds$ is the normalizing constant. Then, the following equality holds:*

$$-D_{\text{KL}}(d^\pi(s;g)\|p(s;g)) + C = (1-\gamma)J(\pi) + \mathcal{H}(d^\pi(s;g)) \quad (54)$$

*where $J(\pi)$ is the GCRL objective (Eq. (2)) with reward $r(s; g)$ and $C := \mathbb{E}_{g\sim p(g)}[\log Z(g)]$ is a constant.*

*Proof.* We have that

$$\begin{aligned}
&(1-\gamma)J(\pi) \\
&= \mathbb{E}_{g\sim p(g)}\mathbb{E}_{s\sim d^\pi(s;g)}[r(s;g)] \\
&= \mathbb{E}_{g\sim p(g)}\mathbb{E}_{s\sim d^\pi(s;g)}\left[\log e^{r(s;g)}\right] \\
&= \mathbb{E}_{g\sim p(g)}\mathbb{E}_{s\sim d^\pi(s;g)}\left[\log \frac{e^{r(s;g)}Z(g)}{Z(g)}\right] \\
&= \mathbb{E}_{g\sim p(g)}\mathbb{E}_{s\sim d^\pi(s;g)}\left[\log \frac{e^{r(s;g)}}{Z(g)}\right] + \mathbb{E}_{g\sim p(g)}[\log Z(g)] \\
&= \mathbb{E}_{g\sim p(g)}\mathbb{E}_{s\sim d^\pi(s;g)}\left[\log \frac{e^{r(s;g)}}{Z(g)} \cdot \frac{d^\pi(s;g)}{d^\pi(s;g)}\right] + C \\
&= \mathbb{E}_{g\sim p(g)}\mathbb{E}_{s\sim d^\pi(s;g)}\left[\log \frac{p(s;g)}{d^\pi(s;g)}\right] + \mathbb{E}_{g\sim p(g)}\mathbb{E}_{d^\pi(s;g)}[\log d^\pi(s;g)] + C \\
&= \mathbb{E}_{g\sim p(g)}\left[-D_{\text{KL}}(d^\pi(s;g)\|p(s;g)) - \mathcal{H}(d^\pi(s;g))\right] + C
\end{aligned} \quad (55)$$

Rearranging the inequality gives the desired result. $\square$

A constant term $C$ appear in the equality to account for the need for normalizing $e^{r(s;g)}$ to make it a proper distribution. This, however, does not change the optimal solution for the goal-conditioned state-occupancy matching objective. Therefore, we have shown that for any choice of reward $r(s;g)$, solving the GCRL problem with a maximum state-entropy regularization is equivalent to optimizing for the goal-conditioned state-occupancy matching objective with target distribution $p(s;g) := \frac{e^{r(s;g)}}{Z(g)}$.

### E.2 Optimality Conditions for Hindsight Relabeling

In section 4.2, we have stated that HER is not optimal for most choices of reward functions. In this section, we investigate conditions under which hindsight relabeling methods such as HER would be optimal.

Let the goal-relabeling distribution for HER be $p_{\text{HER}}(g \mid s, a)$; we do not specify the functional form of $p_{\text{HER}}(g \mid s, a)$ for generality (see [26]). Then, in order for this distribution to be optimal, then it must satisfy

$$p_{\text{HER}}(g \mid s, a) = p(g \mid s, a)(f'_\star(R(s;g) + \gamma \mathcal{T} V^*(s, a; g) - V^*(s; g)) \tag{56}$$

Then, the choice of $r(s;g)$ such that this equality holds is the reward function for which HER would be optimal. However, solving for $r(s;g)$ is generally challenging and we leave it to future work for investigating whether doing so is possible for general $f$-divergence coupled with neural networks.

This optimality condition is related to a prior work [10], which has found that hindsight relabeling is optimal in the sense of maximum-entropy inverse RL [50] for a peculiar choice of reward function (see Equation 9 in [10]), which cannot be implemented in practice. Our result is more general as it applies to any choice of $f$-divergence, and is not restricted to the form of maximum-entropy inverse RL.

### E.3 Theoretical Comparison to Prior Regression-based GCRL methods

In section 4.3, we have stated that GoFAR's theoretical guarantee (Theorem 4.1) is stronger in nature compared to prior regression-based GCRL methods. Here, we provide an in-depth discussion.

Both GCSL [14] and WGCSL [47] prove that their objectives are lower bounds of the true RL objective (Theorem 3.1 in [14] and Theorem 1 in [47], respectively); however, in both works, the lower bounds are loose due to constant terms that do not depend on the policy and hence do not vanish to zero. In contrast, GoFAR's objective (6) is, by construction, a lower bound on the RL objective, as it simply incorporates a $f$-divergence regularization. If the offline data $d^O$ is *on-policy*, then our lower bound is an equality. In contrast, even with on-policy data, the lower bounds in both GCSL and WGCSL are still loose due to the unavoidable constant terms.

GCSL also proves a sub-optimality guarantee (Theorem 3.2 in [14]) under the assumption of full state-space coverage. Though full state-space coverage has been considered in some prior offline RL works [24, 31], it is much stronger than the concentrability assumption in our Theorem 4.1, which only applies to $d^*$. Furthermore, this guarantee is not statistical in nature, and instead directly makes a strong assumption on the *maximum* total-variance distance between $\pi$ and optimal $\pi^*$ for the GCSL objective, which is difficult to verify in practice. In contrast, our bound suggests asymptotic optimality: given enough offline data, the solution to GoFAR's policy objective will converge to $\pi^*$. Finally, WGCSL proves a policy improvement guarantee (Proposition 1 in [47]) under their exponentially weighted advantage; the improvement is not a strict equality, and consequently there is no convergence guarantee to the optimal policy. Furthermore, this result is not directly dependent on their use of an advantage function, so it is not clear the precise role of their advantage function in their algorithm.

## F  Task Descriptions

In this section, we describe the tasks in our experiments in Section 5.

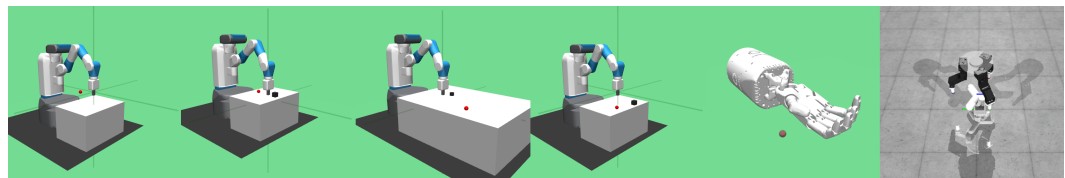

Figure 7: Tasks from left to right: FetchReach, FetchPush, FetchSlide, FetchPick, HandReach, D'Claw (Simulation)

## F.1 Fetch Tasks

The Fetch environments involve the Fetch robot with the following specifications, developed by Plappert et al. [39].

- Seven degrees of freedom
- Two pronged parallel gripper
- Three dimensional goal representing Cartesian coordinates of target
- Sparse, binary reward signal: 0 when at goal with tolerance 5cm and -1 otherwisejk
- 25 Hertz simulation frequency
- Four dimensional action space
    - Three Cartesian dimensions
    - One dimension to control gripper

### F.1.1 Fetch Reach

The task is to place the end effector at the target goal position. Observations consist of the end effector's positional state, whether the gripper is closed, and the end effector's velocity. The reward is given by:

$$r(s, a, g) = 1 - \mathbb{1}(\|s_{xyz,\text{ee}} - g_{xyz}\|_2 \leq 0.05)$$

### F.1.2 Fetch Push

The task is to push an object to the target goal position. Observations consist of the end effector's position, velocity, and gripper state as well as the object's position, rotational orientation, linear velocity, and angular velocity. The reward is given by:

$$r(s, a, g) = 1 - \mathbb{1}(\|s_{xyz,\text{obj}} - g_{xyz}\|_2 \leq 0.05)$$

### F.1.3 Fetch Slide

In this task, the goal position lies outside of the robot's reach and the robot must slide the puck-like object across the table to the goal. Observations consist of the end effector's position, velocity, and gripper state as well as the object's position, rotational orientation, linear velocity, and angular velocity. The reward is given by:

$$r(s, a, g) = 1 - \mathbb{1}(\|s_{xyz,\text{obj}} - g_{xyz}\|_2 \leq 0.05)$$

### F.1.4 Fetch Pick

The task is to grasp the object and hold it at the goal, which could be on or above the table. Observations consist of the end effector's position, velocity, and gripper state as well as the object's position, rotational orientation, linear velocity, and angular velocity. The reward is given by:

$$r(s, a, g) = 1 - \mathbb{1}(\|s_{xyz,\text{obj}} - g_{xyz}\|_2 \leq 0.05)$$

### F.2 Hand Reach

Uses a 24 DoF robot hand with a 20 dimensional action space. Observations consist of each of the 24 joints' positions and velocities. The goal space is 15 dimensional corresponding to the positions of each of its five fingers. The goal is achieved when the mean distance of the fingers to their goals is less than 1cm. The reward is binary and sparse: 0 if the goal is reached and -1 otherwise, i.e.

$$r(s, a, g) = 1 - \mathbb{1}\left(\frac{1}{5}\sum_{i=1}^{5}\|s_i - g_i\|_2 \leq 0.01\right)$$

### F.3 D'ClawTurn (Simulation)

First introduced by Ahn et al. [2], the D'Claw environment has a 9 DoF three-fingered robotic hand. The turn task consists of turning the valve to a desired angle. The initial angle is randomly chosen from $[-\frac{\pi}{3}, \frac{\pi}{3}]$; the target angle is randomly chosen from $[-\frac{2*\pi}{3}, \frac{2*\pi}{3}]$. The observation space is 21D, consisting of the current joint angles $\theta_t$, their velocities $\dot{\boldsymbol{\theta}}$, angle between current and goal angle, and the previous action. The environment terminates after 80 steps. The reward function is defined as:

$$r = \mathbb{1}\left(\left|\arctan2\left(\frac{s_{y,obj}}{s_{x,obj}}\right) - \arctan2\left(\frac{g_{y,obj}}{g_{x,obj}}\right)\right| \leq 0.1\right)$$

### F.4 D'ClawTurn (Real)

To make real-world data collection easier, we slightly modify the initial and target angle distributions. The initial angle is randomly chosen from $[-\frac{\pi}{3}, \frac{\pi}{3}]$; the target angle is randomly chosen from $[-\frac{\pi}{2}, \frac{\pi}{2}]$. Using this task distribution, collecting 400K transitions with random actions takes about 15 hours. In Figure 8, we also include a larger picture of the robot platform.

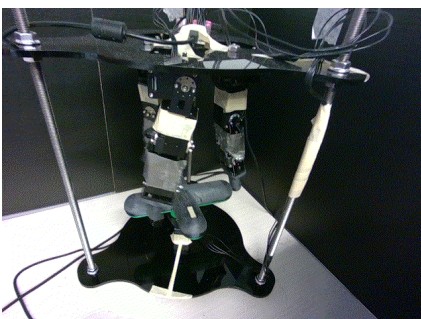

Figure 8: The D'Claw tri-finger platform.

## G   Experimental Details

In this section, we provide experimental details omitted in Section 5 of the main text. These include (1) technical details of the baseline methods, (2) hyperparameter and architecture details for all methods, (3) offline GCRL dataset details, and finally, (4) experimental details of the zero-shot transfer experiment.

### G.1   Baseline Implementation Details

**DDPG.** We use an open-source implementation of DDPG, which has already tuned DDPG on the set of Fetch tasks. We implement all other methods on top of this implementation, keeping identical architectures and hyperparameters when appropriate. The critic objective is

$$\min_Q \mathbb{E}_{(s_t, a_t, s_{t+1}, g) \sim d^{\bar{O}}}[(r(s_t, g) + \gamma\bar{Q}(s_{t+1}, \pi(s_{t+1}, g), g) - Q(s_t, a_t, g))^2] \tag{57}$$

where $\bar{Q}$ denotes the stop-gradient operation. The policy objective is

$$\min_\pi -\mathbb{E}_{(s_t, a_t, s_{t+1}, g) \sim d^{\bar{O}}}[Q(s_t, \pi(s_t, g), g)] \tag{58}$$

DDPG updates the critic and the policy in an alternating fashion.

**ActionableModel.** We implement AM on top of DDPG. Specifically, we add a CQL loss in the critic update:

$$\mathbb{E}_{(s,g)\sim d^{\tilde{O}},a\sim\exp(Q)}[Q(s,a,g)] \tag{59}$$

where $d^{\tilde{O}}$ is the distribution of the relabelled dataset. In practice, we sample $10$ random actions from the action-space to approximate this expectation. Furthermore, we implement goal-chaining, where for half of the relabeled transitions in each minibatch update, the relabelled goals are randomly sampled from the offline dataset. We found goal-chaining to not be stable in some environments, in particular, FetchPush, FetchPickAndPlace, and FetchSlide. Therefore, to obtain better results, we remove goal-chaining for these environments in our experiments.

**GCSL.** We implement GCSL by removing the DDPG critic component and changing the policy loss to maximum likelihood:

$$\min_{\pi} -\mathbb{E}_{(s,a,g)\sim d^{\tilde{O}}}\left[\log \pi(a \mid s, g)\right] \tag{60}$$

**WGCSL.** We implement WGCSL on top of GCSL by including a Q-function. The Q-function is trained using TD error as in DDPG and provided an advantage weighting in the regression loss. The advantage term we compute is $A(s_t, a_t, g) = r(s_t; g) + \gamma Q(s_{t+1}, \pi(s_{t+1}, g); g) - Q(s_t, a_t; g)$. Using this, the WGCSL policy objective is

$$\min_{\pi} -\mathbb{E}_{(s_t,a_t,\phi(s_i))\sim d^{\tilde{O}}}\left[\gamma^{i-t}\exp_{clip}(A(s_t, a_t, \phi(s_i)))\log \pi(a_t \mid s_t, \phi(s_i))\right] \tag{61}$$

where we clip $\exp(\cdot)$ for numerical stability. The original WGCSL uses different HER rates for the critic and the actor training. To make the implementation simple and consistent with all other approaches, we use the same HER rate for both components. We note that the original WGCSL computes the advantage term slightly differently as $A(s_t, a_t, g) = r(s_t; g) + \gamma Q(s_{t+1}, \pi(s_{t+1}, g); g) - Q(s_t, \pi(s_t, g); g)$; this version of WGCSL[3] is incorporated in our open-sourced code.

With the exception of AM, all baselines set the goal-relabeling distribution $d^{\tilde{O}}$ to be the uniform distribution over future states in the same trajectory (See Equation (26)).

### G.2 Architectures and Hyperparameters

Each algorithm uses their own set of fixed hyperparameters for all tasks. WGCSL, GCSL, and DDPG are already tuned on our set of tasks [39, 47], so we use the reported values from prior works; AM, in our implementation, shares same networks as DDPG, so we use DDPG's values. For GoFAR, we use identical hyperparameters as WGCSL because they share similar network components; GoFAR additionally trains a discriminator, for which we use the same architecture and learning rate as the value network. We impose a small discriminator gradient penalty [16] to prevent overfitting. For all experiments, We train each method for 3 seeds, and each training run uses 400k minibatch updates of size 512. The architectures and hyperparameters for all methods are reported in Table 3.

### G.3 Offline GCRL Experiments

**Datasets.** For each environment, the offline dataset composition is determined by whether data collected by random actions provides sufficient coverage of the desired goal distribution. For FetchReach and D'ClawTurn, we find this to be the case and choose the offline dataset to be 1 million random transitions. For the other four tasks, random data does not capture meaningful goals, so we create a mixture dataset with 100K transitions from a trained DDPG-HER agent and 900K random transitions; the transitions are not labeled with their sources. This mixture setup has been considered in prior works [21, 32] and is reminiscent of real-world datasets, where only a small portion of the dataset is task-relevant but all transitions provide useful information about the environment.

### G.4 Zero-Shot Transfer Experiments

We use GoFAR (Binary) variant for trainning the GoFAR planner. The low-level controller is trained using an online DDPG algorithm on a narrow goal distribution, set to be closed to the object's initial positions.

---

[3]We thank Joey Hejna for pointing out this difference in an email correspondence.

Table 3: Offline GCRL Hyperparameters.

|  | Hyperparameter | Value |
|---|---|---|
| Hyperparameters | Optimizer | Adam [22] |
|  | Critic learning rate | 5e-4 (1e-3 for AM/DDPG) |
|  | Actor learning rate | 5e-4 (1e-3 for AM/DDPG) |
|  | Discriminator learning rate | 5e-4 |
|  | Discriminator gradient penalty | 0.01 |
|  | Mini-batch size | 256 |
|  | Discount factor | 0.98 |
| Architecture | Discriminator hidden dim | 256 |
|  | Discriminator hidden layers | 2 |
|  | Discriminator activation function | ReLU |
|  | Critic (resp. Value) hidden dim | 256 |
|  | Critic (resp. Value) hidden layers | 2 |
|  | Critic (resp.Value) activation function | ReLU |
|  | Actor hidden dim | 256 |
|  | Actor hidden layers | 2 |
|  | Actor activation function | ReLU |

GoFAR Hierarchical Controller operates by first generating a sequence of subgoals $(g_1, ..., g_T)$ using $\pi_{\text{high}}$ by recursively feeding the newest generated goal and conditioning on the final goal $g$. Then, at each time step $t$, the low-level controller executes action $\pi_{\text{low}}(a_t \mid s_t, g_t)$. The high-level subgoals are not re-planned during low-level controller execution. We note that this is a simple planning algorithm, and improvement in performance can be expected by considering more sophisticated planning approaches.

# H   Additional Results

## H.1   Offline GCRL Full Results

In this section, we provide the full results table for discounted return, final distance, and success rate metrics, including error bars over 10 random seeds. The number inside the parenthesis indicates the best HER rate for the baseline methods on the task. Star ($\star$) indicate statistically significant improvement over the second best-performing method under a 2-sample $t$-test.

Table 4: Discounted Return on offline GCRL tasks, averaged over 10 random seeds.

| Task | Supervised Learning | | | Actor-Critic | |
|---|---|---|---|---|---|
|  | **GoFAR** (Ours) | WGCSL | GCSL | AM | DDPG |
| FetchReach | $28.2 \pm 0.61$ | $21.9 \pm 2.13 (1.0)$ | $20.91 \pm 2.78 (1.0)$ | $\mathbf{30.1} \pm 0.32 (0.5)$ | $29.8 \pm 0.59 (0.2)$ |
| FetchPick | $\mathbf{19.7} \pm 2.57$ | $9.84 \pm 2.58 (1.0)$ | $8.94 \pm 3.09 (1.0)$ | $18.4 \pm 3.51 (0.5)$ | $16.8 \pm 3.10 (0.5)$ |
| FetchPush ($\star$) | $\mathbf{18.2} \pm 3.00$ | $14.7 \pm 2.65 (1.0)$ | $13.4 \pm 3.02 (1.0)$ | $14.0 \pm 2.81 (0.5)$ | $12.5 \pm 4.93 (0.5)$ |
| FetchSlide | $2.47 \pm 1.44$ | $\mathbf{2.73} \pm 1.64 (1.0)$ | $1.75 \pm 1.3 (1.0)$ | $1.46 \pm 1.38 (0.5)$ | $1.08 \pm 1.35 (0.5)$ |
| HandReach ($\star$) | $\mathbf{11.5} \pm 5.26$ | $5.97 \pm 4.81 (1.0)$ | $1.37 \pm 2.21 (1.0)$ | $0. \pm 0.0 (0.5)$ | $0.81 \pm 1.73 (0.5)$ |
| D'ClawTurn ($\star$) | $\mathbf{9.34} \pm 3.15$ | $0.0 \pm 0.0 (1.0)$ | $0.0 \pm 0.0 (1.0)$ | $2.82 \pm 1.71 (1.0)$ | $0.0 \pm 0.0 (0.2)$ |
| Average Rank | **1.5** | 3 | 4.17 | 2.83 | 4 |

Table 5: Final Distance on offline GCRL tasks, averaged over 10 random seeds.

| Task | Supervised Learning | | | Actor-Critic | |
|---|---|---|---|---|---|
|  | **GoFAR** (Ours) | WGCSL | GCSL | AM | DDPG |
| FetchReach | $0.018 \pm 0.003$ | $0.007 \pm 0.0043 (1.0)$ | $0.008 \pm 0.008 (1.0)$ | $\mathbf{0.007} \pm 0.001 (0.5)$ | $0.041 \pm 0.005 (0.2)$ |
| FetchPickAndPlace | $\mathbf{0.036} \pm 0.013$ | $0.094 \pm 0.043 (1.0)$ | $0.108 \pm 0.060 (1.0)$ | $0.040 \pm 0.020 (0.5)$ | $0.043 \pm 0.021 (0.5)$ |
| FetchPush | $\mathbf{0.033} \pm 0.008$ | $0.041 \pm 0.020 (1.0)$ | $0.042 \pm 0.018 (1.0)$ | $0.070 \pm 0.039 (0.5)$ | $0.060 \pm 0.026 (0.5)$ |
| FetchSlide ($\star$) | $\mathbf{0.120} \pm 0.02$ | $0.173 \pm 0.04 (1.0)$ | $0.204 \pm 0.051 (1.0)$ | $0.198 \pm 0.059 (0.5)$ | $0.353 \pm 0.248 (0.5)$ |
| HandReach ($\star$) | $\mathbf{0.024} \pm 0.009$ | $0.035 \pm 0.012 (1.0)$ | $0.038 \pm 0.013 (1.0)$ | $0.037 \pm 0.004 (0.5)$ | $0.038 \pm 0.013 (0.5)$ |
| D'ClawTurn ($\star$) | $\mathbf{0.92} \pm 0.28$ | $1.49 \pm 0.26 (1.0)$ | $1.54 \pm 0.15 (1.0)$ | $1.28 \pm 0.26 (1.0)$ | $1.54 \pm 0.13 (0.2)$ |
| Average Rank | **1.5** | 2.33 | 4.25 | 2.67 | 4.5 |

Table 6: Success Rate on offline GCRL tasks, averaged over 10 random seeds.

| Task | Supervised Learning | | | Actor-Critic | |
|------|---------------------|---|---|--------------|---|
| | **GoFAR** (Ours) | WGCSL | GCSL | AM | DDPG |
| FetchReach | **1.0** $\pm$ 0.0 | 0.99 $\pm$ 0.01 (1.0) | 0.98 $\pm$ 0.05 (1.0) | **1.0** $\pm$ 0.0 (0.5) | 0.99 $\pm$ 0.02 (0.2) |
| FetchPickAndPlace | **0.84** $\pm$ 0.12 | 0.54 $\pm$ 0.16 (1.0) | 0.54 $\pm$ 0.20 (1.0) | 0.78 $\pm$ 0.15 (0.5) | 0.81 $\pm$ 0.13 (0.5) |
| FetchPush ($\star$) | **0.88** $\pm$ 0.09 | 0.76 $\pm$ 0.12 (1.0) | 0.72 $\pm$ 0.15 (1.0) | 0.67 $\pm$ 0.14 (0.5) | 0.65 $\pm$ 0.18 (0.5) |
| FetchSlide | **0.18** $\pm$ 0.12 | **0.18** $\pm$ 0.14 (1.0) | 0.17 $\pm$ 0.13 (1.0) | 0.11 $\pm$ 0.09 (0.5) | 0.08 $\pm$ 0.11 (0.5) |
| HandReach | **0.40** $\pm$ 0.20 | 0.25 $\pm$ 0.23 (1.0) | 0.047 $\pm$ 0.10 (1.0) | 0.0 $\pm$ 0.0 (0.5) | 0.023 $\pm$ 0.054 (0.5) |
| D'ClawTurn ($\star$) | **0.26** $\pm$ 0.13 | 0.0 $\pm$ 0.0 (1.0) | 0.0 $\pm$ 0.0 (1.0) | 0.13 $\pm$ 0.14 (1.0) | 0.01 $\pm$ 0.02 (0.2) |
| Average Rank | **1** | 3 | 4 | 3.33 | 3.67 |

## H.2  Ablations

We also include the full task-breakdown table of GoFAR ablations presented in Figure 4 for completeness. As shown in 7, GoFAR and GoFAR (HER) perform comparatively on all tasks. GoFAR (binary) is slightly worse across tasks, and GoFAR (KL) collapses due to the use of an unstable $f$-divergence.

Table 7: GoFAR Ablation Studies

| Variants | FetchReach | FetchPickAndPlace | FetchPush | FetchSlide | HandReach | DClawTurn |
|----------|-----------|-------------------|-----------|------------|-----------|-----------|
| GoFAR | 27.8 $\pm$ 0.55 | 19.5 $\pm$ 4.13 | 18.9 $\pm$ 3.87 | 3.67 $\pm$ 0.78 | 11.9 $\pm$ 3.00 | 9.34 $\pm$ 3.15 |
| GoFAR (HER) | 28.3 $\pm$ 0.65 | 19.8 $\pm$ 2.82 | 20.5 $\pm$ 2.29 | 3.85 $\pm$ 0.80 | 8.02 $\pm$ 5.70 | 10.51 $\pm$ 3.51 |
| GoFAR (Binary) | 26.1 $\pm$ 1.14 | 17.4 $\pm$ 1.78 | 17.4 $\pm$ 2.67 | 3.69 $\pm$ 1.75 | 6.01 $\pm$ 1.62 | 5.13 $\pm$ 4.05 |
| GoFAR (KL) | 0 $\pm$ 0.0 | 0 $\pm$ 0.0 | 0 $\pm$ 0.0 | 0 $\pm$ 0.0 | 0 $\pm$ 0.0 | 0 $\pm$ 0.0 |

## H.3  Real-World Dexterous Manipulations

In our qualitative analysis, we visualize all methods on a specific task instance of turning the valve prong (marked by the red strip) clockwise for 90 degree; the goal location is marked by the green strip. The robot initial pose is randomized. As shown in Figure 9, GoFAR reaches the goal with three random initial poses, whereas all baselines fail. See the figure captions for detail. Policy videos are included in the supplementary material.

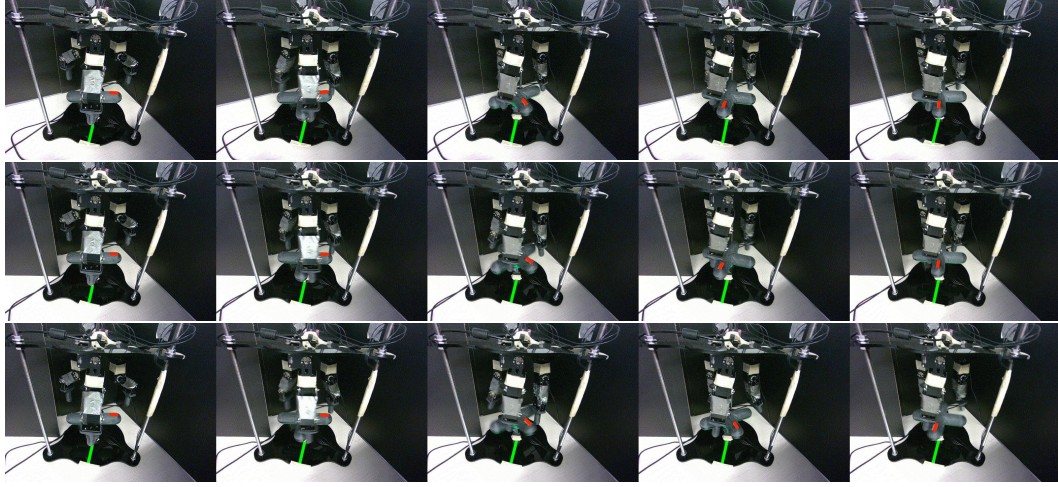

(a) GoFAR robustly achieved the goal with three random initial poses; in the first two runs, it demonstrates "recovery" behavior, as the robot would initially overshoot and then turn the valve counterclockwise. In the last run, the robot initially undershoots and then turns again to reach the goal.

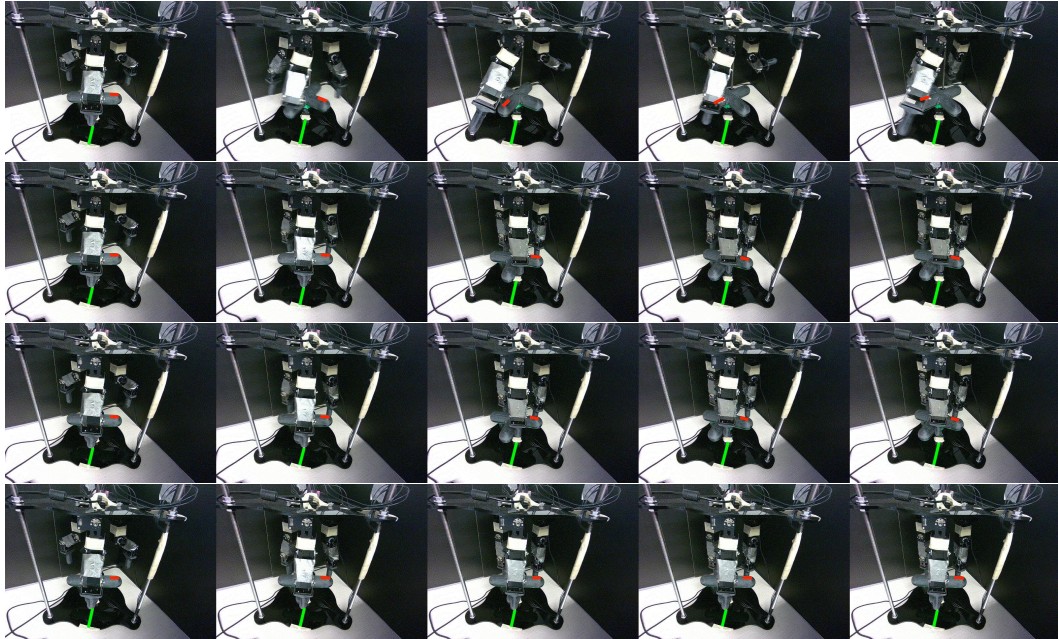

(b) Baselines fail to turn the volve prong (marked by the red strip) to the goal angle (marked by the green strip). AM is the only method that is able to rotate the prong to some degree, though it overshoots in this case and exhibits unnatural behavior.

Figure 9: D'ClawTurn policy visualization.

## H.4 Zero-Shot Plan Transfer

We visualize GoFAR hierarchical controller and the plain low-level controller on three distinct goals in Figure 10. See the figure caption for detail. Policy videos are included in the supplementary material.

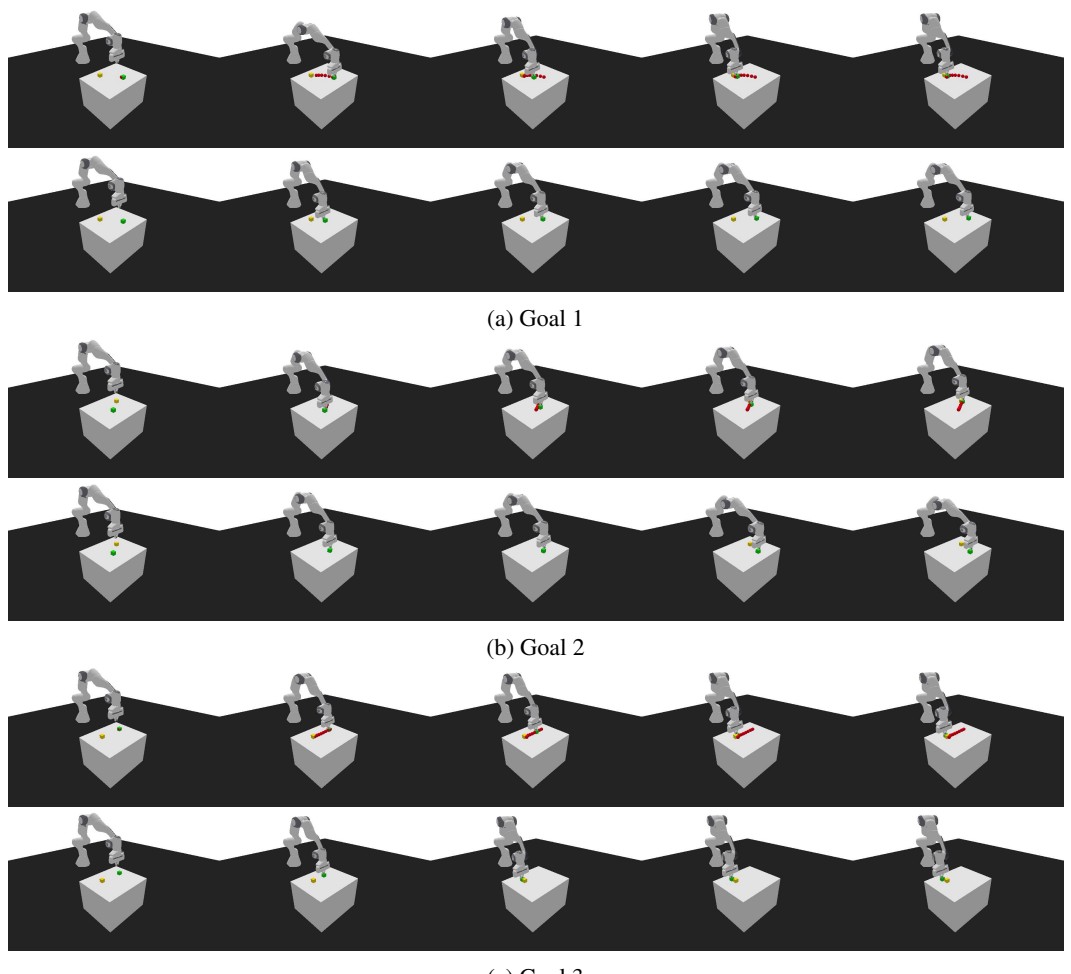

(a) Goal 1

(b) Goal 2

(c) Goal 3

Figure 10: Qualitative comparison of GoFAR hierarchical controller (top) vs. plain low-level controller (bottom) on representative goals in the Franka pushing task. Red circles represent intermediate subgoals generated by the GoFAR planner. As shown, the low-level controller only succeeds in Goal 3, whereas the hierarchical controller achieves the distant goals in all three cases.