# OpenReview forum: "Offline Goal-Conditioned Reinforcement Learning via $f$-Advantage Regression"
_NeurIPS.cc/2022/Conference — NeurIPS 2022 Accept_

### Official Review · Reviewer_D96G · 2022-07-11

**Rating:** 8
**Confidence:** 4
**Soundness:** 4 excellent
**Presentation:** 3 good
**Contribution:** 3 good

**Summary:**

The authors propose a mechanism for performing offline multi-goal reinforcement learning without hindsight goal relabelling, by attempting to mimic the state-occupancy distribution of an agent that teleports directly to the goal. This avoids HER's well-known hindsight bias, and allows GoFAR to perform better on stochastic environments that are challenging for methods that employ hindsight goal relabelling. They find that this method outperforms offline versions of DDPG with HER and several other offline GCRL algorithms on several experiments, including real robot experiments. Additionally, they use the same method to derive a goal-based planner that allows for zero shot transfer to systems with new dynamics, as long as they share the same goal space and reward structure. They additionally prove several finite-sample guarantees for the method.

**Questions:**


For the fetch robot results, I'd like to see training curves shown with error bars for statistical significance. Statistical significance is reported for these experiments, but only in the appendix, which hides that fact that the improvement over other methods is not statistically significant for several tasks. The lack of improvement on these tasks is not in and of itself a problem, but it should be easier for a casual reader to see what is significant. I'd also like to see success rates for these methods -- it's more informative and intuitive than reward alone.

I'd also like to see the goal-conditioned planner elaborated on a bit more, if space allows. For instance, what does a policy over goals mean exactly, and what does it look like? How do the formal guarantees given earlier in the paper translate to a policy that outputs goals

**Limitations:**

There seem to be no significant negative social impacts, and no obvious limitations other than those that are inherited from GCRL and offline RL.

**Strengths And Weaknesses:**

$\textbf{Originality: }$
The method is fairly original. It uses similar algorithm to SMODICE, but adapts it to a goal-conditioned setting. The two most unique insights here seem to be
1) given a goal, the "optimal" teleporting agent's behavior is known $\textit{a priori}$, which means no expert demonstration is needed -- the learning objective can be derived analytically, while still retaining the strong theoretical guarantees from imitation learning.
2) the derived algorithm can be transformed into a planner over intermediate goals. This allows a formulation that does not depend on actions, but instead plans over the goal space, allowing zero-shot generalization to novel dynamics, as long as the goal-space and task remain the same.
These are both interesting and creative insights.

$\textbf{Quality: }$
The math results generally look solid, and provide some nice guarantees on convergence.

The experimental results are slightly more mixed. For several experiments, including the standard fetch experiments, no measures of statistical significance are reported in the main paper, and are only shown in the appendix. Additionally, success rates are not reported for these experiments. Broadly, the results for the Fetch environments  indicate that the method compares favorably against other offline methods, but still falls significantly short of the performance achieved by online methods. A real-robot experiment shows that the method is able to learn a task in which it manipulates a knob, while other methods fail.

Most interesting is the goal planner experiment, in which the authors are able to significantly improve the performance of a weak policy by learning a sequence of goals that may be zero-shot transferred across different domains. They find that using the planer to guide the weak policy significantly improves the success rate.

$\textbf{Clarity: }$
Although the math can be a tad difficult to get through in places, I found it easier to follow than many other DICE-based papers. Broadly speaking, the core ideas are clear and well-articulated.

$\textbf{Significance: }$
The proposed method solves the bias problem in the offline setting and provides valuable convergence guarantees. Additionally it provides good experimental results and proposes an interesting new approach to zero shot cross-domain transfer. The one downside is that it's a bit niche, by virtue of sitting at the intersection of two subfields, goal conditioned RL and offline RL.

---

> ### Author Response · Authors · 2022-08-02
> **Response to Reviewer D96G**
>
> We thank the reviewer for their constructive comments and positive assessment of our work! Here, we respond to the questions and comments the reviewer raises. Please let us know if we can provide any additional clarifications during the discussion period.
>
> ---
> **Question 1**: How statistically significant are the reported results? Does GoFAR provide statistically significant improvement over baselines?
>
> **Response 1**:
> We have included the error-bar version of Table 1 (reproduced below) in the main text and have run additional 7 random seeds (to make it 10 random seeds in total for each experiment) to further improve the robustness of our results. As shown, GoFAR attains the best overall performance across six tasks, and the results are statistically significant on three tasks (tasks with (\*) next to their name), including the two more challenging  dexterous manipulation tasks. Furthermore, we would like to highlight that GoFAR’s improvement to the next best regression-based method WGCSL is statistically significant on five tasks.
>
> |      Task      | Supervised Learning |                 |              |   Actor-Critic  |             |
> |:--------------:|:-------------------:|:---------------:|:------------:|:---------------:|:-----------:|
> |                |     GoFAR (Ours)    |      WGCSL      |     GCSL     |        AM       |     DDPG    |
> |   FetchReach   |     28.2 ± 0.61     |   21.9 ± 2.13   | 20.91 ± 2.78 | **30.1** ± 0.32 | 29.8 ± 0.59 |
> |    FetchPick   |   **19.7** ± 2.57   |   9.84 ± 2.58   |  8.94 ± 3.09 |   18.4 ± 3.51   | 16.8 ± 3.10 |
> |  FetchPush(\*) |   **18.2** ± 3.00   |   14.7 ± 2.65   |  13.4 ± 3.02 |   14.0 ± 2.81   | 12.5 ± 4.93 |
> |   FetchSlide   |     2.47 ± 1.44     | **2.73** ± 1.64 |  1.75 ± 1.30 |   1.46 ± 1.38   | 1.08 ± 1.35 |
> |  HandReach(\*) |   **11.5** ± 5.26   |   5.97 ± 4.81   |  1.37 ± 2.21 |    0.0 ± 0.0    | 0.81 ± 1.73 |
> | D'ClawTurn(\*) |   **9.34** ± 3.15   |    0.0 ± 0.0    |   0.0 ± 0.0  |   2.82 ± 1.71   |  0.0 ± 0.0  |
> |  Average Rank  |       **1.5**       |        3        |     4.17     |       2.83      |      4      |
>
> ---
> **Question 2**: Can success rates be reported for the experiments?
>
> **Response 2**:
>
> We have included the success rate table in Appendix H.1 of our revised manuscript; it is reproduced below for your convenience. As shown, GoFAR is consistently among the best methods on all tasks. Including this metric, we have shown that GoFAR is consistently the best performing method on three distinct metrics of relative merits and often leads to statistically significant improvement on the more difficult tasks.
>
> |                | Supervised Learning |                 |              |  Actor-Critic |               |
> |:--------------:|:-------------------:|:---------------:|:------------:|:-------------:|:-------------:|
> |                |     GoFAR (Ours)    |      WGCSL      |     GCSL     |       AM      |      DDPG     |
> |   FetchReach   |    **1.0** ± 0.0    |   0.99 ± 0.01   |  0.98 ± 0.05 | **1.0** ± 0.0 |  0.99 ± 0.02  |
> |    FetchPick   |   **0.84** ± 0.12   |   0.54 ± 0.16   |  0.54 ± 0.20 |  0.78 ± 0.15  |  0.81 ± 0.13  |
> |  FetchPush(\*) |   **0.88** ± 0.09   |   0.76 ± 0.12   |  0.72 ± 0.15 |  0.67 ± 0.14  |  0.65 ± 0.18  |
> |   FetchSlide   |   **0.18** ± 0.12   | **0.18** ± 0.14 |  0.17 ± 0.13 |  0.11 ± 0.09  |  0.08 ± 0.11  |
> |  HandReach(\*) |   **0.40** ± 0.20   |   0.25 ± 0.23   | 0.047 ± 0.10 |   0.0 ± 0.0   | 0.023 ± 0.054 |
> | D'ClawTurn(\*) |   **0.26** ± 0.13   |    0.0 ± 0.0    |   0.0 ± 0.0  |  0.13 ± 0.14  |  0.01 ± 0.02  |
> |  Average Rank  |        **1**        |        3        |       4      |      3.33     |      3.67     |
>
> ---
> **Question 3**: In the goal-conditioned planner setting, what is a policy over goals? Does the convergence guarantee of Theorem 4.1 extend to this setting?
>
> **Response 3**: A policy over goals would output a state instead of an action; its implementation is identical to an usual action policy, except that its output dimension would now match the goal dimension. A real-world analogy of this would be that a goal policy would output the next street to navigate to, conditioned on the current location (state) and target location (goal), whereas an action policy for the same task would output the exact steering command.
> The formal guarantee directly transfers over to this setting because we are now mimicking the **optimal** goal-conditioned planner; in practice, this means replacing every occurrence of $a$ in Theorem 1 with next state $s’$.
>
> We thank the reviewer again for their time and effort helping us improve our paper!

---

> > ### Comment · Reviewer_D96G · 2022-08-05
> > **Response to Paper7863 Authors**
> >
> > Thank you for your elaborations on the experiments and your clarifications. I will upgrade my assessment accordingly.
> >
> > I still am somewhat confused about the goal-conditioned planner however. Is this a distribution over exclusively the next state? Or is it a discounted average over future trajectories like a successor representation?
> > More importantly, what does it mean for such a distribution to be optimal or suboptimal? For a policy over actions, optimality means that sampling actions according to the policy yields cumulative rewards greater than or equal to the rewards you would get by sampling from any other policy. However, it's not immediately clear to me how sampling these goals yields a distribution over rewards, because you still need a way of generating actions from these goals to get a set of valid trajectories. Is the idea that, if you have an optimal goal conditioned planner $\pi^*_G(\phi(s') \mid \phi(s), G)$ and an optimal policy $\pi^*_A(a \mid s, G)$, then sampling a goal $g$ from $\pi^*_G(\phi(s') \mid \phi(s), G$ and then an action from $\pi^*_A(a \mid s, g)$ still gives you an optimal distribution over actions? Or is there a sense in which intermediate goals can be treated as actions such that $\pi^*_G$ is optimal in this new space?

---

> > > ### Author Response · Authors · 2022-08-07
> > > **Follow-Up Response to Reviewer D96G**
> > >
> > > We thank the reviewer for their response and improved assessment of our work! Here, we address the reviewer’s remaining questions about our goal-conditioned planning setup. Please let us know if you have any remaining questions, and we are happy to provide further clarifications.
> > >
> > > Recall that reward functions in GCRL are typically defined over just states and goals (Line 120 in the paper), $r(s;g)$. Therefore, sampling intermediate states does yield a distribution over rewards, and the optimal goal-conditioned planner is the one that yields cumulative rewards greater than or equal to the rewards you would get by sampling from any other goal-conditioned planner. As the reviewer suggested, this amounts to treating intermediate goals as actions and the notion of optimality is indeed defined in this new action space.
> > >
> > > Now, as the reviewer suggested, the optimal planner would still require a lower-level action policy to generate valid trajectories in the environment. And when the action policy is itself optimal, then the overall combination would indeed be optimal.
> > >
> > > The advantage of learning an optimal goal-conditioned planner using our algorithm is that this planner is agnostic to the underlying action policy used to navigate to intermediate goals. This enables transferring the learned plan to a new embodiment with a different action space. As different robots, despite their action space differences, would use the same high-level plan to solve the same task, GoFAR’s zero-shot plan transfer capability relieves the need of learning a globally optimal action policy for a new robot and instead only requires an action policy that can reach nearby goals. We have demonstrated this use case in Section 5.4, in which the goal-conditioned planner learned using offline Fetch robot data, is able to **double** the success rate of a pre-existing action policy in the target Franka robot domain by guiding this action policy to optimal intermediate goals required to reach a distant final goal.

---

### Official Review · Reviewer_gkGs · 2022-07-11

**Rating:** 5
**Confidence:** 4
**Soundness:** 2 fair
**Presentation:** 3 good
**Contribution:** 3 good

**Summary:**

This paper proposes a new regression-based offline GCRL algorithm from the perspective of state-occupancy matching. Compared to previous methods, GoFAR does not require any hindsight relabeling. GoFAR also decouples the training process of value and policy updating. These features make GoFAR enjoy the benefits of better performance, training stability, and zero-shot transfer to new target domains.

**Questions:**

1. Regarding the first concern listed above, why we can choose $r(s; g) = p(s; g)$ in the general offline RL setting? If not, can we still cast offline goal-conditioned RL as a state-occupancy matching problem?
2. Can Proposition 1 be derived from the aspect of variational inference? It seems that Proposition 1 can be derived from variational inference with fewer lemmas and assumptions.
3. Rewards signal: What’s the difference between discriminator-based reward and dataset-based reward? Could you add some ablations experiments using these two types of reward in the model?


**Limitations:**

The main limitation of this work is stated in my concerns regarding the reformulation of offline goal-conditioned RL as an imitation learning problem. If the authors can provide convincing responses to the above questions (especially Question 1), I’d like to raise my score.

**Strengths And Weaknesses:**

The biggest contribution of this paper is casting offline goal-conditioned RL as a state-occupancy matching problem. After this formulation, GoFAR reuses techniques from existing offline imitation work [1][2] (see below) to solve the state-occupancy matching problem by regarding the data that accomplish the desired goal as expert data, and the whole offline data as supplemental data.
However, I do have several concerns about this reformulation of offline goal-conditioned RL.

1. The objective of goal-conditioned RL is to find a policy that maximizes the cumulative reward, while state-occupancy matching is only mimicking the expert. Suppose: 1) the reward is not sparse reward (not the case that only gets 1 reward when accomplishing the goal) and 2) the dataset contains successful but not optimal trajectories that accomplish the goal g (i.e., the cumulative reward is low). In this case, mimicking these trajectories will only get suboptimal policies.
2. In Appendix E.1, the authors give a theoretical connection between Goal-Conditioned State-Matching and Probabilistic GCRL, however, this builds on the premise that we can choose $r(s; g) = p(s; g)$. Why is it a valid choice under RL (i.e., have the freedom to choose $r$ as $p$)? It is reasonable under the imitation learning setting but is problematic under the RL setting. As the reward function is given by the problem and $p(s;g)$ is given by the offline data.

Aside from the concerns, the additional strengths and weaknesses of the paper are listed as follows

**Strengths:**

* This paper provides a novel view to solve the offline GCRL problem by adopting state-occupancy matching. It avoids hindsight relabeling, improves training stability, and owns the zero-transfer ability.
* The authors connect well with probabilistic GCRL and HER.
* The proposed method performs well on offline GCRL tasks and the authors conducted comprehensive experiments to show the advantage of GoFAR.

**Weaknesses:**
* The learning objective of GoFAR, Eq. (5), is a lower bound of $KL(d^{\pi}, p)$, rather than the exact KL divergence, and objective (7) is an even looser lower bound.
* The benchmark experiments are not sufficient. Could you test your algorithm on the offline GCRL benchmark introduced in [3]?

[1] Kim et al, DemoDICE: Offline Imitation Learning with Supplementary Imperfect Demonstrations. ICLR, 2022.

[2] Ma et al, Smodice: Versatile offline imitation learning via state occupancy matching. ICML, 2022.

[3] Yang et al, Rethinking Goal-conditioned Supervised Learning and Its Connection to Offline RL. ICLR, 2022.

---

> ### Author Response · Authors · 2022-08-02
> **Response to Reviewer gkGs (Part 1)**
>
> We thank the reviewer for their thoughtful comments and suggestions! Here, we respond to the  questions and comments the reviewer raises. Please let us know if you have lingering questions and whether we can provide any additional clarifications during the discussion period to improve the score.
>
> ---
> **Question/Concern 1**: Can we cast goal-conditioned RL as goal-conditioned state-occupancy matching for any reward function $r(s;g)$?
>
> **Response 1**: Yes, we can always cast offline goal-conditioned RL as a goal-conditioned state-occupancy matching problem. We have re-written Appendix E.1 as well as the beginning of Section 4 (Proposition 4.1) to make this result clear. We reproduce this proposition here for your convenience:
>
> **Proposition 4.1**:
> Given any choice of $r(s;g)$, for each $g$ in the support of $p(g)$, define $p(s;g) = \frac{e^{r(s;g)}}{Z(g)}$, where $Z(g) := \int e^{r(s;g)} ds$ is the normalizing constant. Then, the following equality holds:
> \begin{equation}
> -D_{KL}(d^\pi(s;g) \| p(s;g)) + C = (1-\gamma)J(\pi) + \mathcal{H}(d^\pi(s;g))
> \end{equation}
>
> where $J(\pi)$ is the GCRL objective with reward $r(s;g)$ and $C := \mathbb{E}_{g \sim p(g)}[\log Z(g)]$.
>
>
> At a high level, this proposition says that for any choice of reward $r(s;g)$, solving the GCRL problem while maximizing the state-entropy is equivalent to optimizing for the goal-conditioned state-occupancy matching objective with target distribution $p(s;g) := \frac{e^{r(s;g)}}{Z(g)}$.
>
> We acknowledge that the original wording of “choosing $p(s;g)=r(s;g)$” in our proposition E.1 is ambiguous and can be construed as changing a problem-provided reward function to this choice in order to make the formulation work out. But as explained above, that is not what our theory is suggesting to do, and the proposition is merely shedding light on the one-to-one correspondence between GCRL and goal-conditioned state-occupancy matching, which provides a principled justification for using the latter as a solution approach to the former.
>
> ---
> **Question/Concern 2**: The objective of goal-conditioned RL is to find a policy that maximizes the cumulative reward, while state-occupancy matching is only mimicking the expert. Mimicking successful, but suboptimal trajectories are not optimal when the reward function is not sparse.
>
> **Response 2**: As stated in our first response, this characterization of state-occupancy matching is incorrect. Maximizing cumulative rewards can be formulated as a state-occupancy matching problem for general reward functions. We agree that mimicking sub-optimal goal-reaching trajectories will not be optimal, but emphasize that this is not what our method GoFAR does. GoFAR mimics the **optimal** goal-conditioned policy for Eq. (8) under any choice of reward r(s;g); this is formally derived in section 4.2, Eq. (16). Our Figure 1 also captures this idea as there are multiple trajectories that lead to the highlighted goal, but GoFAR would learn to mimic the most optimal ones.
>
> Now, as the reviewer suggested, if the reward is not sparse and provided, then we can directly use this reward in Eq. (8) and onward, and GoFAR will learn to mimic the optimal policy for this reward.  The issue of mimicking sub-optimal goal-reaching trajectories that the reviewer describes is rather more characteristic of prior GCRL methods, such as GCSL, which treats each trajectory as optimal for the goal it reached and performs behavior cloning on each trajectory conditioned on its achieved goal.
>
> This distinction is best illustrated in our real-world dexterous manipulation experiment (Section 5.3), for which the offline dataset is collected by a random policy. Here, the dataset indeed contains successful trajectories that are highly sub-optimal (e.g., trajectories that randomly stumble upon a desired goal knob configuration). In this challenging setting, GoFAR discovers useful sub-trajectories to mimic in order to learn a globally coherent, competent policy, whereas baselines such as GCRL fail to make any progress as they are designed to mimic these sub-optimal trajectories as they are, which are not effective strategies for the task.

---

> > ### Author Response · Authors · 2022-08-02
> > **Response to Reviewer gkGs (Part 2)**
> >
> > ---
> > **Question/Concern 3**: What’s the difference between discriminator-based reward and dataset-based reward? Are there ablation results for this?
> >
> > **Response 3**: The discriminator-based reward is a learned reward function by training a classifier that distinguishes states drawn from the derived (or user-specified in the case that a reward function is not given) $p(s;g)$ and the empirical state distribution in the offline data $d^O(s;g)$; using this reward gives the tightest lower bound in Eq. (5) and is necessary when the offline dataset does not come with rewards. On the other hand, when the dataset comes with sparse rewards, then, we can leverage the relationship that $r(s;g) \propto \log p(s;g)$ to directly use the dataset provided rewards. Now, we can again train a discriminator as before and optimize for the tighter lower bound. However, doing so requires training an additional neural network, and for simplicity, we may just use the provided reward as it is, corresponding to optimizing the looser lower bound Eq. (7). We have provided this ablation in Figure 3 in the original submission, where the discriminator-based reward version of GoFAR is denoted as GoFAR, and the dataset-based reward is denoted as GoFAR (Binary). As suggested by our theory, GoFAR obtains better results, but GoFAR (binary) is simpler to implement and is still superior to all baselines. The flexibility of GoFAR with its choice of reward is a strength as it gives practitioners a choice in optimizing over performance vs. simplicity.
> >
> > ---
> > **Question/Concern 4**: Can Proposition 4.2 (originally Proposition 4.1) be derived from the perspective of variational inference?
> >
> > **Response 4**: We believe that Proposition 4.2 is different from variational inference (VI). In VI, an unobserved latent variable is introduced to obtain a tractable lower bound to the observed data likelihood. In Proposition 4.2, states, actions, and goals are all observed, and the lower bound is a direct result of the definition of state-occupancy and its relation to state-action occupancy (Lemma B.1).
> >
> > ---
> > **Question/Concern 5**: The learning objectives of GoFAR (Eq. (5) and (7)) are lower bounds of the goal-conditioned state-occupancy matching objective. This is a weakness of the method.
> >
> > **Response 5**:
> > Optimizing a lower bound to an otherwise difficult, often intractable, objective should not be considered a weakness itself. This strategy has been highly successful in machine learning (e.g., evidence lower bound (ELBO) in variational inference) and has been adopted by prior offline state-occupancy matching works [1][2], though in different settings to ours.
> > In GoFAR’s case, the exact KL divergence cannot be easily optimized in the offline setting because it is an expectation over $d^\pi$ and hence requires sampling from policy $\pi$, which cannot be performed in the offline setting (Line 141-144). We derive our lower-bound objective from first principle and have shown how it leads to stable and effective offline GCRL both in theory (Section 4.3) and practice (Section 5.1-5.3).
> >
> > ---
> > **Question/Concern 6**: The experiments should include the benchmark from [3].
> >
> > **Response 6**: The first five tasks in Section 5.1 (FetchReach, FetchPick, FetchPush, FetchSlide, HandReach) and their datasets are directly taken from [3]; we have made this clear in our revised version of the paper. These five tasks are the “harder” tasks as stated in [3], and we did not include the results for the “easier” tasks because we found all methods to perform comparably, a finding supported by the results in [3] (see g-BC and HER in Table 1 of [3]). Instead, we include an experiment with a real robot of high-dimensional action space, on which we showed significant gain compared to the baselines. As offline RL is a paradigm motivated by real-world challenges of reinforcement learning, we believe our real-world results make a stronger case of our algorithm’s practical utility than results on additional simulated environments of low-dimensional inputs (i.e., the tasks we did not include from [3]).
> >
> > ---
> > [1] Kim et al, DemoDICE: Offline Imitation Learning with Supplementary Imperfect Demonstrations. ICLR, 2022.
> >
> > [2] Ma et al, Smodice: Versatile offline imitation learning via state occupancy matching. ICML, 2022.
> >
> > [3] Yang et al, Rethinking Goal-conditioned Supervised Learning and Its Connection to Offline RL. ICLR, 2022.
> >
> > We thank the reviewer again for their time and effort helping us improve our paper! Please let us know if we can provide additional clarifications to improve our score.

---

> > > ### Comment · Reviewer_gkGs · 2022-08-07
> > > **Response to the authors**
> > >
> > > Thank you for clarifying the comments I raised. Some of my concerns have been addressed. However, I still have some reservations about this paper regarding to the choice of $r(s;g)$ used for derivation and the similarity to SMODICE. Specifically:
> > > - In Appendix E.1, the equation developed in Proposition E.1 (also 4.1 in the main text) is under the choice of $r(s;g)=p(s;g)$, then how can you still have the freedom to choose $p(s;g)=e^{r(s;g)}$ in order to make the connection with the original GCRL objective? This change of choice on $r(s;g)$ seems vital in the proposed method, otherwise it is not possible to make the connection between $(1-\gamma)J(\pi)= E_{g\sim p(g), s\sim d^{\pi}(s;g)}[r(s;g)]\approx \log E_{g\sim p(g), s\sim d^{\pi}(s;g)}[p(s;g)]$. I’m still not quite convinced by the argument provided by the author.
> > > - I suspect that the authors are actually solving a related, non-linear version of the objective $J(\pi)$, rather than the original exact form of $J(\pi)$ in Eq.(2). As this proposition plays a foundational role in the later development of the algorithm, I would like the authors to further justify the validity of this treatment.
> > > - As the derivation of the model is largely built upon SMODICE but adds additional goal-conditioned setting, I feel that it inherited too much imitation learning flavor rather than RL. The differences and innovations over SMODICE should be clearly stated or discussed in the paper.

---

> > > > ### Author Response · Authors · 2022-08-07
> > > > **Follow-Up Response to Reviewer gkGs**
> > > >
> > > > Thank you for your response! We have provided responses to your concerns; please let us know if there are any remaining questions.
> > > >
> > > > **Clarification on the connection between $r$ and $p$ and on Proposition 4.1.**
> > > >
> > > > We assume the reviewer is referring to the original Propositions E.1. This original proposition provided an *alternative* way to connect $r$ and $p$, *not* a change of choice. Defining $p(s;g)=e^{r(s;g)}/Z(g)$ is the simplest and most natural way to obtain our result. To avoid confusion, we have revised our paper to exclusively use the choice $p(s;g)=e^{r(s;g)}/Z(g)$. Our updated Proposition E.1 is now also included in the main text as Proposition 4.1 (reproduced in our first response).
> > > >
> > > > To be precise, the flow of ideas is now the following:
> > > >
> > > > * The environment comes with a reward function $r(s;g)$
> > > > * Our algorithm defines $p(s;g)=e^{r(s;g)}/Z(g)$
> > > > * Our algorithm uses this choice of $p(s;g)$ in the goal-conditioned state-occupancy matching objective it optimizes to solve the original goal-conditioned RL (GCRL) problem
> > > >
> > > > Then, Proposition 4.1 says that for *any* $r(s;g)$, the original GCRL objective (with the addition of a state-entropy maximization term) is equivalent to the objective optimized by our algorithm. This updated Proposition also makes it clear that GoFAR is optimizing the original GCRL objective, not a non-linear transformation of it.
> > > >
> > > > Finally, we emphasize that all reward-based evaluation metrics (Discounted Return in Table 1, Success Rate in Table 6 in Appendix H.1) are presented using the true reward from the original environment, demonstrating that our approach successfully solves the original goal-conditioned RL problem.
> > > >
> > > >
> > > > **GoFAR’s Differences and Contributions over SMODICE.**
> > > >
> > > >  Here, we summarize both the conceptual and algorithmic innovations of GoFAR.
> > > >
> > > > Our conceptual contribution is demonstrating that goal-conditioned state-occupancy matching can be used to solve GCRL; this critical connection justifies extending SMODICE to GoFAR in the first place. Furthermore, we show that GoFAR carries numerous properties (Section 4.2-4.4) that are not present in the SMODICE paper and not possessed by any other offline GCRL algorithm we compare to. Given GoFAR’s novelty with respect to prior offline GCRL algorithms as well as its strong theoretical foundation and empirical performance, we believe that these conceptual contributions already make GoFAR a significant contribution to the offline GCRL literature.
> > > >
> > > > Algorithmically, GoFAR also differs from SMODICE. For instance, a naive extension of SMODICE to the GCRL setting would require having access to a separate dataset of demonstrations from an optimal goal-conditioned policy (the expert dataset $d^E$ in SMODICE), which is a strong assumption not satisfied by the standard GCRL problem setting. GoFAR avoids the need for this dataset by training a discriminator to distinguish states from goals (in contrast, the discriminator in SMODICE is trained to distinguish states in the offline dataset vs. states in the expert dataset). Our derivation of GoFAR demonstrates that this alternative discriminator is sufficient to express the original GCRL objective.
> > > >
> > > > In addition, GoFAR can be trained without having to train a discriminator, which is required by SMODICE. We derive this fact mathematically in Eq. (8), and experimentally in Section 5.1. We have shown that this variant, referred to as GoFAR (Binary) in our experiments, still outperforms all baselines. Discriminator training in adversarial (imitation) learning is known to be prone to overfitting and instability; removing this requirement makes GoFAR far easier to apply in practice.
> > > >
> > > > We have updated our related work section to accentuate our paper’s contributions relative to SMODICE (highlighted in red in updated Section 2).
> > > >
> > > > Finally, we do not believe that “[inheriting] too much imitation learning flavor” is undesirable, especially given GoFAR’s stronger theoretical guarantees and algorithmic properties (Section 4.2-4.4), and its strong empirical performance over all baselines across all settings we have tested (Section 5.1-5.4). Additionally, we note that many current state-of-art algorithms in the vanilla offline RL setting are heavily influenced by ideas from imitation learning, e.g., TD3-BC [2], Decision-Transformer [3], or IQL [4].
> > > >
> > > > [1] Ma et al, Smodice: Versatile offline imitation learning via state occupancy matching. ICML, 2022.
> > > >
> > > > [2] Fujimoto et al., A minimalist approach to offline reinforcement learning. NeurIPS, 2021.
> > > >
> > > > [3] Chen et al., Decision transformer: Reinforcement learning via sequence modeling. NeurIPS 2021.
> > > >
> > > > [4] Kostrikov et al., Offline Reinforcement Learning with Implicit Q-Learning. ICLR, 2022.
> > > >
> > > >
> > > > We thank the reviewer again for their time and thoughtful feedback. Please let us know if our new responses have addressed your remaining concerns.

---

> > > > > ### Comment · Reviewer_gkGs · 2022-08-08
> > > > > **Response**
> > > > >
> > > > > Thank you for the further clarification on $r$ and $p$. Now it seems technically more solid in this part. I have raised my score accordingly.

---

### Official Review · Reviewer_U8M8 · 2022-07-12

**Rating:** 8
**Confidence:** 4
**Soundness:** 4 excellent
**Presentation:** 3 good
**Contribution:** 3 good

**Summary:**

The paper presents an approach for offline goal-conditioned RL that is based on state-occupancy matching. In particular, the paper outlines a procedure for a goal-conditioned state occupancy optimization in an offline setting by deriving a lower bound on the optimization objective such that it does not require online sampling. While doing so, the algorithm incorporates f-divergence based dual optimization and decouples value function from actor policy training, which improves stability of training. In contrast to prior works, the method does not require hindsight relabeling, which further stabilizes the training. The algorithm is compared to multiple recent GCRL baselines and it is shown to outperform them on several simulated robotic manipulation tasks as well as in a dexterous real-world D’Claw environment.

**Questions:**

Please refer to Strengths And Weaknesses for a list of questions.

**Limitations:**

The authors discussed limitations of their work. The proposed approach does not lead to any obvious negative societal impacts.

**Strengths And Weaknesses:**

The paper presents an approach for offline goal-conditioned RL that is based on state-occupancy matching. In particular, the paper outlines a procedure for a goal-conditioned state occupancy optimization in an offline setting by deriving a lower bound on the optimization objective such that it does not require online sampling. While doing so, the algorithm incorporates f-divergence based dual optimization and decouples value function from actor policy training, which improves stability of training. In contrast to prior works, the method does not require hindsight relabeling, which further stabilizes the training. The algorithm is compared to multiple recent GCRL baselines and it is shown to outperform them on several simulated robotic manipulation tasks as well as in a dexterous real-world D’Claw environment.

Strengths
- Efficiently training goal-conditioned policies from offline data promises to enable training on large and diverse robotic datasets.
- The paper is well-written, and straightforward to understand and follow. Mathematical derivations are coherent.
- The method is shown to work on a variety of interesting manipulation robotic tasks, including a real world dexterous manipulation task.
- The paper provides source code of the method implementation.

Criticism
- It would be interesting to see an evaluation of the method on higher-dimensional and especially articulated visual goal tasks to better understand the scalability of the approach.
- It would be interesting to see an evaluation on tasks that require stitching of multiple episodes to accomplish a task, to better understand the ability of the method to combine experiences from different trajectories.

---

> ### Author Response · Authors · 2022-08-02
> **Response to Reviewer U8M8**
>
> We thank the reviewer for their constructive comments and positive assessment of our work! Here, we respond to the questions and comments the reviewer raises. Please let us know if we can provide any additional clarifications during the discussion period.
>
> ---
> **Question/Comment 1**: It would be interesting to scale GoFAR to visual tasks.
>
> **Response 1**: We agree that this would be an interesting future work, and we do not foresee major technical barriers as several offline RL algorithms have been successfully applied to visual tasks. Given that GoFAR is more stable than the baselines, many of which have already been applied to visual goal tasks (e.g., AM and DDPG-HER), we expect GoFAR to be able to scale to visually complex goal-reaching tasks.
>
> ---
> **Question/Comment 2**: It would be interesting to test GoFAR on tasks that require trajectory “stitching”.
>
> **Response 2**: We agree that this is an interesting experimental setting and argue that our real-robot experiment already showcases GoFAR’s trajectory stitching capability. Note that GoFAR’s value training (Eq. (9)) should already enable sub-trajectory stitching capability because it is performing approximate dynamic programming. We indeed see that this is the case in our real-world dexterous manipulation experiment (Section 5.3), for which the offline dataset is collected by a random policy. Here, the dataset contains mostly trajectories that are highly sub-optimal (e.g., trajectories that randomly stumble upon a desired goal knob configuration). Successfully solving the task requires the learning algorithm to be able to stitch together useful sub-trajectories to learn a more effective strategy for solving the task. This is precisely what GoFAR’s combination of temporal-difference based value learning and regression-based policy learning is designed to do. As shown in our paper and supplementary videos, GoFAR discovers useful sub-trajectories to mimic in order to learn a globally coherent, competent policy, whereas baselines such as GCRL fail to make any progress as they are designed to mimic these sub-optimal trajectories as they are, which are not effective strategies for the task.
>
> We thank the reviewer again for their time and effort helping us improve our paper!

---

> > ### Comment · Reviewer_U8M8 · 2022-08-07
> > **Thanks for your response**
> >
> > Thank you for clarifications! Given my initial high rating and concerns of other reviewers, I would like to keep my rating.

---

### Official Review · Reviewer_PYXz · 2022-07-18

**Rating:** 7
**Confidence:** 4
**Soundness:** 3 good
**Presentation:** 3 good
**Contribution:** 3 good

**Summary:**

This work presents a method for *offline* goal-conditioned policy learning, from offline data that *does not contain actions or rewards*, using only a specified target goal distribution p(g). The approach, named GoFAR, is formulated as a state-occupancy matching objective, and draws upon a similar derivation as SMODICE [1], with additional modifications to be applicable to the goal-conditioned setting. As a result, GoFAR inherits the advantages of SMODICE. Experiments study various components of the GoFAR method, provide real-world robotic results, and demonstrate cross-embodiment transfer.

References:

[1] Versatile Offline Imitation from Observations and Examples via Regularized State-Occupancy Matching


**Questions:**

- Why is this called goal-conditioned RL? The objective being discussed throughout the paper, i.e. objective (1), is a state-occupancy matching objective that does not have any rewards involved. Also the title does not reflect that the method is directed towards the setting where the offline dataset contains states only (no rewards or actions).
- I don't see how the teleportation described in the abstract is relevant to this work? Do you mean because p(g) is a distribution of goal states?
- I think in this work the authors need to discuss SMODICE [1] more thoroughly as the method is very closely related. Furthermore, a number of "advantages" of GoFAR that are listed are due to the use of SMODICE-like methodology, rather than a direct contribution of this work. Such aspects should be more clearly discussed. Also in Section 4.1, it would be good if the manuscript noted that the derivations draw upon similar derivations in prior DICE lines of work.
- For state-occupancy matching related work it would be good to cite [2] as they also present results with state-marginal/occupancy matching, and present results for target state distributions that were not generated by an agent (e.g. hand designed target state distributions).
- Equation (7): It is not clear to me why this is a looser bound than (5). Do you have a derivation for why this would be a lowerbound to (5)?
- Equation (11): You should explain why you are proposing to learn the policy using equation 11. Namely the fact that the "f-advantage" is equal (at least theoretically) to the density ratio $\frac{d^*}{d^O}$.
- Line 192: Can you please elaborate on how Bayes rule leads to that equality? Also, I don't have an intuition for what p(g | s, a) is? You explained initially that p(g) is a "desired goal distribution", but I don't understand what p(g | s, a) would mean?
- Section 5.1: It would be good to note which methods are state-only, which methods are state and action, and which methods use rewards. Despite the appendix, it would also be good to at least briefly mention what are the task rewards (e.g. dense or sparse), how the goal distributions are defined, etc.
- Section 5.1: Instead of DDPG, why did you not opt for an offline RL method, given that online RL algorithms tend to perform poorly in offline settings?
- Section 5.1: Please report standard deviation of results, even if it's only 3 random seeds. It is very challenging to understand the significance of your results without error metrics.
- Section 5.1: Can you elaborate on how the sparse binary reward is used with equation (7) to implement GoFAR (binary)?
- Baselines: I think a potentially unfair advantage that GoFAR has compared to the other approaches is that it is being given $p(g)$, the distribution of desired goal states. In contrast for example, HER is relabeling any future state as a goal state and performing updates using all those possible relabeld "goals". I think important alternative baselines would be: 1) use $p(g)$ in HER as well. For example, only do HER relabeling using goals that are in the support of $p(g)$. 2) Perform goal-conditioned offline RL, e.g. using CQL or any favorite offline RL method, where the input to the policy / value function is (s, g) / (s, a, g), where g ~ p(g), and the reward is relabeled as r(s, a, g).
- In the appendix G you don't describe how the $\tilde{O}$ relabeled distribution is obtained for each method, e.g. GCSL.
- Section 5.2: Without errorbars, it is not possible to evaluate the significance of the results.

With regards to aspects that would directly help me better evaluate and improve my score:
- Addressing my concern regarding baselines, either verbally convincing me that I am mistaken in my thought process / assumptions, or implementing the two suggested baselines and demonstrating that they would not resolve the poorer performance of the baselines you tried.
- Highlighting exactly what you believe are the core contributions of *your work*, not aspects of your work that are due to the use of SMODICE, etc. What can GoFAR do that prior methods cannot (e.g. type of data, type of available information, etc.)? What is the first impressive result you would show a person? If I am choosing an algorithm for my use-case, why would I opt for GoFAR as opposed to the other methods (currently I am not convinced solely by numerical results since there is a lack of reporting of standard deviation / error bars across random seeds, as well as my concern regarding baselines discussed above)?

References:

[1] Versatile Offline Imitation from Observations and Examples via Regularized State-Occupancy Matching

[2] A Divergence Minimization Perspective on Imitation Learning Methods



**Limitations:**

Authors have not highlighted the limitations / potential failure cases / future work for GoFAR.

**Strengths And Weaknesses:**

Originality:
- I would not necessarily consider this work very original. As the authors point out, goal-conditioned reinforcement / imitation learning (as well as their state-only variants) are important well-studied problems. Furthermore, the GoFAR approach appears to be an extension of SMODICE [1] to the goal-conditioned setting.

Quality:
- I believe the quality of the paper is good. The writing is largely clear, and the experiments try to cover various aspects of the approach. However, there are a number of questions / concerns noted below.

Clarity:
- The paper is generally well-written

Significance:
- I think the significance can be better understood after authors response to my questions / concerns below.

References:

[1] Versatile Offline Imitation from Observations and Examples via Regularized State-Occupancy Matching

---

> ### Author Response · Authors · 2022-08-02
> **Response to Reviewer PYXz (Part 1)**
>
> We thank the reviewer for their thoughtful comments and suggestions. Here, we respond to the questions and comments the reviewer raises. Please let us know if you have lingering questions and whether we can provide any additional clarifications during the discussion period to improve the score.
>
> ---
> **Question/Comment 1**: What are GoFAR’s core contributions in relation to SMODICE?
>
> **Response 1**: The core technical contributions of GoFAR are (1) showing that GCRL can be solved by goal-conditioned state-occupancy matching, and (2) demonstrating the unique effectiveness of GoFAR’s optimization procedure in GCRL.
>
> GoFAR and SMODICE study different problem settings, and extending the latter to a goal-conditioned RL setting is non-trivial. It is certainly a priori not obvious that offline GCRL can be formulated as a goal-conditioned state-occupancy matching problem. Offline GCRL, unlike offline imitation learning, does not come with a separate expert dataset for task supervision. Instead, proposing to use goals included in the offline dataset as an imaginary “teleporting” expert, introducing the novel goal-conditioned state-occupancy matching objective, and mathematically justifying this intuition (Proposition 4.1 in the updated manuscript) requires technical insights not present in SMODICE.
>
> In addition to demonstrating that SMODICE-like methodology is plausible for offline GCRL, the advantages of GoFAR (Section 4.2: Optimal Goal Weighting, Section 4.3: Uninterleaved Optimization, Section 4.4: Goal-Conditioned Planning) are discovered and proved in this work and not in SMODICE. It is a priori not obvious that this $f$-Advantage Regression procedure they share admits these properties. The first and last properties are emergent only in the goal-conditioned setting and deriving them requires new technical insights that are not readily apparent from SMODICE.
>
> Furthermore, these advantages’ significance should be evaluated in the context of GCRL. Optimal Goal Weighting (Section 4.2) makes GoFAR, to the best of our knowledge, the first GCRL algorithm in the standard sparse-reward GCRL setting that does not depend on some sort of goal relabeling for practical performance. Uninterleaved Optimization (Section 4.3) enables a reduction of offline GCRL to imitation learning and enables deriving a strong convergence guarantee on GoFAR’s performance (Theorem 4.1). This convergence guarantee is also in itself significant in the context of GCRL, as it makes GoFAR the first practical GCRL algorithm with provable finite-sample performance guarantee. Combined with its Goal-Conditioned Planning (Section 4.4) capability, GoFAR is also the first algorithm that can simultaneously learn an optimal goal-conditioned planner.
>
> Together, we believe that establishing an intricate connection between two distinct problems, showing that an existing approach for one can be extended to the other, and demonstrating that the adapted approach is provably effective and enjoys new emergent properties that address fundamental challenges (e.g., dependency on hindsight relabeling, hindsight bias, unstable optimization, etc) in the new setting is a significant contribution.

---

> > ### Author Response · Authors · 2022-08-02
> > **Response to Reviewer PYXz (Part 2)**
> >
> > ---
> > **Question 2**: GoFAR has access to p(g), which is an unfair advantage over baselines. A new baseline that incorporates p(g) in its relabeling distribution should be considered. In addition, a baseline that incorporates offline RL algorithms (e.g., CQL) should be considered.
> >
> > **Response 2**: We clarify that GoFAR is not given $p(g)$ in the sense that GoFAR knows the parametric form of $p(g)$. Rather, GoFAR has access to samples from $p(g)$ through the offline dataset, in which every transition is of the form $(s,a,s’;g)$ where $g$ is sampled from $p(g)$. The baselines have access to the same dataset, and GoFAR does not assume any additional information that is not provided to the baselines. Complete pseudocode for GoFAR is included as Algorithm 2 in Appendix C.3, which details how GoFAR interacts with the static offline dataset during its learning process.
> >
> > Furthermore, we note that the proposed baselines by the reviewer are well-exemplified by our ActionableModel (AM) baseline, which does use samples from $p(g)$ in its relabeling process and performs offline GCRL using a strong offline RL method. In particular, AM first relabels goals from both future states and random goals in the dataset; this can be thought of setting the relabeling distribution as a mixture distribution of the future state distribution and p(g). Then, AM would use CQL to optimize the policy using these relabeled data. In our experiments, we found that incorporating p(g) into the relabeling distribution leads to poor performance on several tasks in our experiments, potentially due to overly conservative estimates for Q(s,a;g) at these goals that impedes policy learning. Hence, for three of our tasks (FetchPush, FetchPickAndPlace, and FetchSlide), we found AM to perform much better with just future states as the relabeling distribution, and this variant of AM is what we reported for these tasks in Table 1 in Section 5.1. AM is competitive as it is the strongest baseline in both our simulation and real-world robot results.
> >
> > Please let us know if this explanation helps clarify our method’s assumptions and the sufficiency of the baselines we already reported in the paper.
> >
> > ---
> > **Question/Concern 3**: What can GoFAR do that prior methods cannot (e.g. type of data, type of available information, etc.)? What is the first impressive result you would show a person? If a practitioner is choosing an offline GCRL for a novel use-case, why should GoFAR be preferred over prior methods?
> >
> > **Response 3**: There are several capabilities that are unique to GoFAR. First, GoFAR does not require the offline dataset to contain reward labels, an assumption that prior works make but is loosened in our work. In addition, GoFAR, as we show in Section 4.4, can also be adapted to learn a goal-conditioned planner using states-only dataset; this is not possible with prior approaches like WGCSL, AM, and DDPG, which learns an action-dependent Q-function that cannot be trained when the dataset is states-only.
> >
> > Beyond these distinct capabilities that we demonstrate in our experiments, the stand-out impressive result we would show a person is our real-world dexterous manipulation results. Recall that this real-robot experiment is extremely challenging because it combines high-dimensional state and action spaces, stochastic dynamics, and highly sub-optimal data all in one. The combination of these factors indeed renders all our baselines ineffective, yet GoFAR still learns an effective policy. We highlight that GoFAR worked “out of box” on this experiment without any additional hyperparameter tuning for this task. On the contrary, all other methods required significant tuning of their hindsight relabeling ratio even on the simulation tasks. And GoFAR’s lack of dependency on hindsight relabeling and improved robustness to stochastic dynamics make it much more likely to “work out of box” on a new (real-world) use-case, and this is precisely the reason that one may opt for GoFAR as opposed to methods.

---

> > > ### Author Response · Authors · 2022-08-02
> > > **Response to Reviewer PYXz (Part 3)**
> > >
> > >
> > > ---
> > > **Question/Concern 4**: The experimental results in Section 5.1 and Section 5.2 need to be substantiated with error bars.
> > >
> > > **Response 4**: We have included error bars for the results in Section 5.1 (the error-bar version was originally in Appendix H.1 for space reason) and 5.2 and have run additional 7 random seeds (to make it 10 random seeds in total for each experiment) to further improve the robustness of our results.
> > >
> > > As shown in the updated Table 1 (reproduced below), GoFAR attains the best overall performance across six tasks, and the results are statistically significant on three tasks, including the two more challenging  dexterous manipulation tasks. Furthermore, we would like to highlight that GoFAR’s improvement to the next best regression-based method WGCSL is statistically significant on five tasks.
> > >
> > > The updated results for Section 5.2 are included in Figure 4 of the main text. We see that there is low intra-seed variance for each method under different levels of stochastic noise (for some methods, the error bar is too small to see given the large y-axis range). Given this low variance, these results indeed support the hypothesis that GoFAR is more robust to stochastic noise than baselines that utilize hindsight goal-relabeling.
> > >
> > > |      Task      | Supervised Learning |                 |              |   Actor-Critic  |             |
> > > |:--------------:|:-------------------:|:---------------:|:------------:|:---------------:|:-----------:|
> > > |                |     GoFAR (Ours)    |      WGCSL      |     GCSL     |        AM       |     DDPG    |
> > > |   FetchReach   |     28.2 ± 0.61     |   21.9 ± 2.13   | 20.91 ± 2.78 | **30.1** ± 0.32 | 29.8 ± 0.59 |
> > > |    FetchPick   |   **19.7** ± 2.57   |   9.84 ± 2.58   |  8.94 ± 3.09 |   18.4 ± 3.51   | 16.8 ± 3.10 |
> > > |  FetchPush(\*) |   **18.2** ± 3.00   |   14.7 ± 2.65   |  13.4 ± 3.02 |   14.0 ± 2.81   | 12.5 ± 4.93 |
> > > |   FetchSlide   |     2.47 ± 1.44     | **2.73** ± 1.64 |  1.75 ± 1.30 |   1.46 ± 1.38   | 1.08 ± 1.35 |
> > > |  HandReach(\*) |   **11.5** ± 5.26   |   5.97 ± 4.81   |  1.37 ± 2.21 |    0.0 ± 0.0    | 0.81 ± 1.73 |
> > > | D'ClawTurn(\*) |   **9.34** ± 3.15   |    0.0 ± 0.0    |   0.0 ± 0.0  |   2.82 ± 1.71   |  0.0 ± 0.0  |
> > > |  Average Rank  |       **1.5**       |        3        |     4.17     |       2.83      |      4      |
> > >
> > > ---
> > > **Question/Concern 5**: The objective being discussed throughout the paper, i.e. objective (1), is a state-occupancy matching objective that does not have any rewards involved. How is it related to goal-conditioned RL? Is GoFAR directed towards the setting where the offline dataset contains states only (no rewards or actions)?
> > >
> > > **Response 5**: (Offline) goal-conditioned RL is the problem we are solving (see Section 3: Problem Formulation). The goal-conditioned state-occupancy matching objective (1) is our proposed solution. In Proposition 4.1 in the updated manuscript as well as Appendix E.1, we show how goal-conditioned state-occupancy matching can be used to solve general offline GCRL problems for general reward functions.
> > >
> > > The main problem setting we study is standard offline GCRL (see Section 3), in which a static offline dataset $D=((s,a,s’,g))$ (reward is optional for our method, but not for our baselines) is given to learn a goal-conditioned policy. In Section 4.4, we show how GoFAR can be used for learning a goal-conditioned planner using states-only datasets. This is an additional and distinct feature of our approach, but not the entire scope of this paper.
> > >
> > > ---
> > > **Question/Comment 6**: How does the teleportation idea in the introduction relate to this paper?
> > >
> > > **Response 6**: The idea of teleportation is relevant to the extent that it helps provide intuition for our method. In state-occupancy matching based imitation learning, the expert distribution is usually generated from running an expert agent in the environment. However, in our setting, the “expert” goal-conditioned state-distribution $p(s;g)$ is merely the distribution of states that satisfy a given goal and is not realizable by any agent in the environment because these goal-satisfying states typically require more than one step of environment interaction to reach. Thus, we can think of this target distribution $p(s;g)$ as the state-distribution of an expert agent who can “teleport” to goal-satisfying states in one step.

---

> > > > ### Author Response · Authors · 2022-08-02
> > > > **Response to Reviewer PYXz (Part 4)**
> > > >
> > > > ---
> > > > **Question/Comment 7**: How is the sparse binary reward used with equation (7) to implement GoFAR (Binary)?
> > > >
> > > > **Response 7**: GoFAR (binary) uses the sparse binary reward provided in the offline dataset $D=\{(s,a,r,s’,g)\}$ as it is, and can be understood as an implementation of Eq. (7) for a particular choice of $p(s;g)$. In Eq. (7), the reward is defined to be $r(s;g) = \log p(s;g)$. If the reward is sparse binary, then this is implicitly encoding that the unnormalized density of $p(s;g)$ is $e$ at $s=g$ and $1$ at all states $s \neq g$. Therefore, GoFAR with sparse binary reward amounts to optimizing Eq. (7) with this choice of $p(s;g)$.
> > > >
> > > > ---
> > > > **Question/Comment 8**: Derivation and intuition for why Eq. (7) is a lower bound to Eq. (5).
> > > >
> > > > **Response 8**: The derivation is included in Appendix B.1. The intuition is that the removed cross-entropy term, $\mathbb{E}_{(s,g)\sim d^\pi(s,g)}[\log \frac{1}{d^O(s;g)}]$, is always positive, so removing it makes (7) a lower bound to (5).
> > > >
> > > > ---
> > > > **Question/Comment 9**: [2] should be cited when discussing related work on state-occupancy matching.
> > > >
> > > > **Response 9**: We have already included this work in our citations (citation [13] in our paper) when referring to prior work in occupancy matching. In our revised manuscript, we have added a reference to this work when discussing matching hand-designed target state distributions. GoFAR differs in that it first demonstrates that GCRL can be formulated as a state-occupancy matching problem and then proposes a tractable optimization procedure in the offline setting.
> > > >
> > > > ---
> > > > **Question/Comment 10**: It would be good to note which methods are state-only, which methods are state and action, and which methods use rewards. Despite the appendix, it would also be good to at least briefly mention what are the task rewards (e.g. dense or sparse), how the goal distributions are defined, etc.
> > > >
> > > > **Response 10**: All methods, including ours, in Section 5.1 use states and actions because they are learning goal-conditioned policies. All baselines use the rewards (i.e., they assume the offline dataset to be labeled with rewards), but GoFAR does not because it trains a discriminator. However, GoFAR can optionally use the rewards and forgo training a discriminator; this variant is tested in our ablations, referred to as GoFAR (Binary).
> > > >
> > > > All task rewards are sparse and the goal distributions are defined as an uniform distribution over valid configurations in either the robot space (robot pose in FetchReach) or object space (e.g., block position in FetchPush). We have included these details under the “Tasks” paragraph in Section 5.1.
> > > >
> > > > ---
> > > > **Question/Comment 11**: The goal-relabeling distribution needs to be specified for the baselines in Appendix G.
> > > >
> > > > **Response 11**: We have included this detail in our revised Appendix G. All baselines, except AM, implement the goal-relabeling distribution as an uniform distribution over future states achieved in the trajectory. We note that this is also the original goal-relabeling distribution used by these baselines.
> > > >
> > > > ---
> > > > **Question/Comment 12**: What does $p(g\mid s,a)$ mean and how does Bayes rule lead to the equality in Eq. (13)?
> > > >
> > > > **Response 12**: $p(g \mid s, a)$ is the distribution of goals conditioned on a state-action pair in the offline data distribution $d^O$. Consider sampling $(s,a,g)$ triplet from the offline dataset. We can first sample a goal from p(g), and then sample (s,a) from the conditional distribution of state-action pair seen in the dataset conditioned on the goal, $d^O(s,a;g)$; this corresponds to the decomposition of $p(g)d^O(s,a;g)$. Alternatively, we can first sample a (s,a) pair from the marginal state-action distribution $d^O(s,a)$, and then sample a goal $g$ that has been seen together with $(s,a)$ in the offline dataset according to the conditional distribution $p(g \mid s,a)$; this corresponds to the decomposition $d^O(s,a)p(g\mid s,a)$.
> > > >
> > > > Now, this equality can be equivalently derived from Bayes rule, which states that $p(g\mid s,a) = \frac{d^O(s,a;g) p(g)}{d^O(s,a)}$. Multiplying both sides by $d^O(s,a)$ gives the desired equality. We have updated the paper to include this more detailed explanation.
> > > >
> > > > ---
> > > > [1] Yang et al, Rethinking Goal-conditioned Supervised Learning and Its Connection to Offline RL. ICLR, 2022.
> > > >
> > > > [2] A Divergence Minimization Perspective on Imitation Learning Methods
> > > >
> > > > We thank the reviewer again for their time and effort helping us improve our paper! Please let us know if we can provide additional clarifications to improve our score.

---

> > > > > ### Comment · Reviewer_PYXz · 2022-08-09
> > > > > **Thank you for your detailed response to my comments, and the updates to your revised pdf**
> > > > >
> > > > > Some new comments:
> > > > > - I don't think I agree with proposition 4.1 being a contribution, as this form is relatively standard in the maximum-entropy RL literature. More specifically, in maxent RL the objective is to obtain a policy whose trajectory distribution is $p(\tau) \sim e^{\sum_t r(s_t, a_t)}$. Formulate as an optimization problem this becomes $D_f(p^\pi(\tau) \vert\vert e^{\sum_t r(s_t, a_t)}/Z)$, and because optimizing over full trajectories is difficult, the formulation is often converted to state-action occupancy matching (as in GAIL, AIRL, etc.). There also exist conditional variants such as [1] and [2].
> > > > > - Proposition 4.2 seems to be an extension of SMODICE Theorem 1.
> > > > > - Proposition 4.3 is an extension of SMODICE section 3.3, which is itself due to prior works such as Section 6.1 of [3].
> > > > > - I agree that section 4.2 provides a nice connection.
> > > > > - The uninterleaved optimization property is also due to SMODICE Algorithm 1.
> > > > > - Section 4.4 is a new contribution.
> > > > > - Algorithm 2: In line 5 how do you sample state and goal pairs? Where is this distribution coming from? In line 6 how do you sample states? And how do these two distributions differ from the ones that your baselines use? Similar question for line 12.
> > > > > - Response 8: I don't think cross-entropy is necessarily positive when the distributions are continuous. Take for example the following cross entropy for two distributions on the real numbers: $p(x) = 1$ for $x \in [0, 1]$, $q(x) = 64$ for $x \in [0, \frac{1}{64}]$, then $-E_{p(x)}[\log q(x)] = -\frac{6}{64}$.
> > > > > - Response 12: I don't follow how you can gain access to $d^O(s, a; g)$? Could you clarify?
> > > > > - There is a mismatch between equation 54, and what you prove in equation 55. You seem to have dropped the expectation over $p(g)$. Same in proposition 4.1.
> > > > >
> > > > > In general, I want to clarify that I am not trying to dismiss the contributions of your work. My point is that in Section 4 it is good to be more transparent about what are novel contributions coming from your work vs. extenions from prior work.
> > > > >
> > > > > Given the clarifications on the contributions of your work, the completeness of empirical results, and the connect made to prior works in this domain, I plan to increase my score to an accept.
> > > > >
> > > > > Best of luck,
> > > > >
> > > > > Reviewer PYXz
> > > > >
> > > > > References:
> > > > > - [1] SMILe: Scalable Meta Inverse Reinforcement Learning through Context-Conditional Policies
> > > > > - [2] Meta-Inverse Reinforcement Learning with Probabilistic Context Variables
> > > > > - [3] Reinforcement learning via fenchel-rockafellar duality

---

> > > > > > ### Author Response · Authors · 2022-08-09
> > > > > > **Thank You for Your Review**
> > > > > >
> > > > > > Dear reviewer PYXz,
> > > > > >
> > > > > > Thank you for your time and thoughtful feedback. We will include the posted references and connections to the mentioned prior works/problem formulations in our next revision. Here, we provide some brief clarifications to the new questions.
> > > > > >
> > > > > > **Question 1: How does sampling work in Line 5,6, and 12 in Algorithm 2?**
> > > > > >
> > > > > > Because our offline data consists of tuples of the form is of the form $(s,a,s',r, g)$, sampling goal-state pairs amount to sampling $(s,a,s',r,g)$ tuples and then only using $s$ and $g$ from the sampled tuples. Sampling from $p(s;g)$ in Line 6 could be implemented by sampling from $s$ in $D$ for which $(s,a,s',r,g)$ has $r=1$.  Sampling from $d^O(s,a;g)$ amounts to sampling state-action pairs from all trajectories that have goal $g$.
> > > > > >
> > > > > > **Question 2: Cross-entropy can be negative for continuous distributions.**
> > > > > >
> > > > > > This is indeed true in general because the (differential) entropy function itself also has this property. For this reason, entropy and cross-entropy functions are typically restricted to random variables taking discrete values. Our mathematical results should also be understood in the context of discrete $S$, $A$, and $G$. We will add this clarification in our next revision.
> > > > > >
> > > > > > **Question 3: $p(g)$ is dropped in Equation (54) and (55).**
> > > > > >
> > > > > > This is a typo;we will correct this error in our next revision.
> > > > > >
> > > > > > We again thank the reviewer for helping us improve our paper!
> > > > > >
> > > > > > Best,
> > > > > >
> > > > > > Paper 7863 Authors

---

### Author Response · Authors · 2022-08-02
**Rebuttal Revision Has Been Posted**

Dear Reviewers, AC, and PC,

Our revised paper has been posted. For clarity, we highlighted all changes in red. The PDF file now contains both the main text as well as the Appendix for ease of reading. The main changes include:

1. Updated Table 1 which includes error bars and is now averaged over 10 random seeds.
2. Updated Appendix E.1 and new Proposition 4.1 that highlights and clarifies the connection between offline GCRL and offline goal-conditioned state-occupancy matching.

We thank all reviewers for their time and effort helping us improve our paper, and we look forward to discussing our paper with you further during the discussion period.

Best,

Paper 7863 Authors

---

> ### Author Response · Authors · 2022-08-07
> **Second Rebuttal Revision Has Been Posted**
>
> Dear Reviewers, AC, and PC,
>
> Our new revised paper has been posted. For clarity, we highlighted all changes in red. The PDF file now contains both the main text as well as the Appendix for ease of reading. The main changes from the first revision include:
>
> 1. Expanded Related Work (Section 2) highlighting GoFAR's differences and contributions with respect to SMODICE.
> 2. Revised and condensed Appendix E.1 to further clarify the relationship between goal-conditioned state-occupancy matching and the original GCRL objective.
>
> We again thank all reviewers for their time and effort helping us improve our paper, and we look forward to continuing discussing our paper with you further during the remaining discussion period.
>
> Best,
>
> Paper 7863 Authors

---

### Meta-Review · Area_Chair_wmkF · 2022-08-29

**Recommendation:** Accept
**Confidence:** Certain

**Metareview:**

The paper presents a novel method for offline goal-conditional RL that is based on reformulating offline Goal-Contitional RL as a state-occupancy matching problem. From this observation, the author are able to leverage and adapt previous work (in particular SMODICE) to their setting. Reviewers were in agreement this was a strong paper, with excellent writing, good mathematical rigor and thorough experimental work. There were some concerns regarding similarity to SMODICE which the authors addressed in their rebuttal.

**Award:**

Yes

---

### Decision · Program_Chairs · 2022-09-14

Accept